# Navigating the Maze of Explainable AI: A Systematic Approach to Evaluating Methods and Metrics

**Lukas Klein** [1,2,3]    **Carsten Lüth** [1,3,4]    **Udo Schlegel** [5]    **Till Bungert** [1,3,4]
**Mennatallah El-Assady** [2]    **Paul Jäger** [1,3]

[1]German Cancer Research Center (DKFZ), Interactive Machine Learning Group, Germany
[2]ETH Zürich, Department of Computer Science, Switzerland
[3]Helmholtz Imaging, German Cancer Research Center (DKFZ), Germany
[4]Heidelberg University, Department of Computer Science, Germany
[5]University of Konstanz, Department of Computer Science, Germany

`lukas.klein@dkfz.de`

## Abstract

Explainable AI (XAI) is a rapidly growing domain with a myriad of proposed methods as well as metrics aiming to evaluate their efficacy. However, current studies are often of limited scope, examining only a handful of XAI methods and ignoring underlying design parameters for performance, such as the model architecture or the nature of input data. Moreover, they often rely on one or a few metrics and neglect thorough validation, increasing the risk of selection bias and ignoring discrepancies among metrics. These shortcomings leave practitioners confused about which method to choose for their problem. In response, we introduce LATEC, a large-scale benchmark that critically evaluates 17 prominent XAI methods using 20 distinct metrics. We systematically incorporate vital design parameters like varied architectures and diverse input modalities, resulting in 7,560 examined combinations. Through LATEC, we showcase the high risk of conflicting metrics leading to unreliable rankings and consequently propose a more robust evaluation scheme. Further, we comprehensively evaluate various XAI methods to assist practitioners in selecting appropriate methods aligning with their needs. Curiously, the emerging top-performing method, Expected Gradients, is not examined in any relevant related study. LATEC reinforces its role in future XAI research by publicly releasing all 326k saliency maps and 378k metric scores as a (meta-)evaluation dataset. The benchmark is hosted at: `https://github.com/IML-DKFZ/latec`.

## 1 Introduction

Explainable AI (XAI) methods have become essential tools in numerous domains, allowing for a better understanding of complex machine learning decisions. The most prevalent XAI methods are saliency maps [58]. As the diversity and abundance of proposed saliency XAI methods expand alongside their growing popularity, ensuring their reliability becomes paramount [2]. Given that there is no clear "ground truth" for individual explanations (e.g., discussed in Adebayo et al. [3]), the trustworthiness of XAI methods is typically determined by examining three key criteria: their accuracy in reflecting a model's reasoning ("faithfulness") [8, 55], their stability under small changes ("robustness") [71, 5], and the understandability of their explanations ("complexity") [12, 10]. Beyond qualitative assessment of saliency maps such as in Doshi-Velez and Kim [21], Ribeiro et al. [50], Shrikumar et al. [57], which can be influenced by human biases and does not scale to large-scale

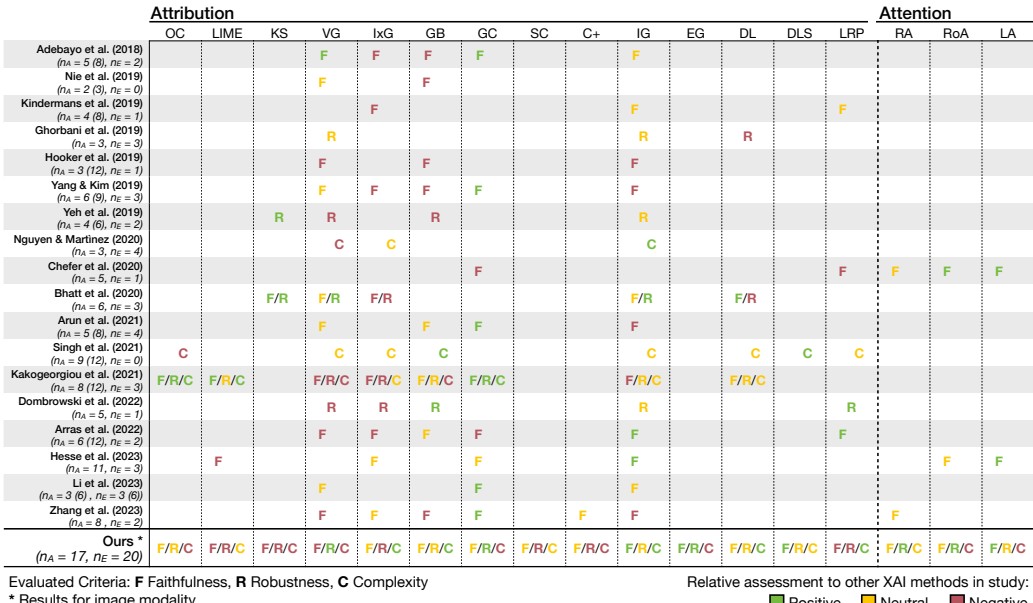

| | Attribution | | | | | | | | | | | | | | | Attention | | |
|---|---|---|---|---|---|---|---|---|---|---|---|---|---|---|---|---|---|---|
| | OC | LIME | KS | VG | IxG | GB | GC | SC | C+ | IG | EG | DL | DLS | LRP | RA | RoA | LA |
| Adebayo et al. (2018) ($n_A$ = 5 (8), $n_E$ = 2) | | | | F | F | F | F | | | F | | | | | | | |
| Nie et al. (2019) ($n_A$ = 2 (3), $n_E$ = 0) | | | | F | | F | | | | | | | | | | | |
| Kindermans et al. (2019) ($n_A$ = 4 (8), $n_E$ = 1) | | | | | F | | | | | F | | | | F | | | |
| Ghorbani et al. (2019) ($n_A$ = 3, $n_E$ = 3) | | | | R | | | | | | R | | R | | | | | |
| Hooker et al. (2019) ($n_A$ = 3 (12), $n_E$ = 1) | | | | F | | F | | | | F | | | | | | | |
| Yang & Kim (2019) ($n_A$ = 6 (9), $n_E$ = 3) | | | | F | F | F | F | | | F | | | | | | | |
| Yeh et al. (2019) ($n_A$ = 4 (6), $n_E$ = 2) | | | R | R | | R | | | | R | | | | | | | |
| Nguyen & Martinez (2020) ($n_A$ = 3, $n_E$ = 4) | | | | C | C | | | | | C | | | | | | | |
| Chefer et al. (2020) ($n_A$ = 5, $n_E$ = 1) | | | | | | | F | | | | | | | F | F | F | F |
| Bhatt et al. (2020) ($n_A$ = 6, $n_E$ = 3) | | | F/R | F/R | F/R | | | | | F/R | | F/R | | | | | |
| Arun et al. (2021) ($n_A$ = 5 (8), $n_E$ = 4) | | | | F | | F | F | | | F | | | | | | | |
| Singh et al. (2021) ($n_A$ = 9 (12), $n_E$ = 0) | C | | | C | C | C | | | | C | | C | C | C | | | |
| Kakogeorgiou et al. (2021) ($n_A$ = 8 (12), $n_E$ = 3) | F/R/C | F/R/C | | F/R/C | F/R/C | F/R/C | F/R/C | | | F/R/C | | F/R/C | | | | | |
| Dombrowski et al. (2022) ($n_A$ = 5, $n_E$ = 1) | | | | R | R | R | | | | R | | | | R | | | |
| Arras et al. (2022) ($n_A$ = 6 (12), $n_E$ = 2) | | | | F | F | F | F | | | F | | | | F | | | |
| Hesse et al. (2023) ($n_A$ = 11, $n_E$ = 3) | | F | | | F | | F | | | F | | | | | | F | F |
| Li et al. (2023) ($n_A$ = 3 (6), $n_E$ = 3 (6)) | | | | F | | | F | | | F | | | | | | | |
| Zhang et al. (2023) ($n_A$ = 8, $n_E$ = 2) | | | | F | F | F | F | | F | F | | | | | F | | |
| Ours * ($n_A$ = 17, $n_E$ = 20) | F/R/C | F/R/C | F/R/C | F/R/C | F/R/C | F/R/C | F/R/C | F/R/C | F/R/C | F/R/C | F/R/C | F/R/C | F/R/C | F/R/C | F/R/C | F/R/C | F/R/C |

Evaluated Criteria: **F** Faithfulness, **R** Robustness, **C** Complexity
\* Results for image modality

Relative assessment to other XAI methods in study:
■ Positive ■ Neutral ■ Negative

Table 1: Showing gaps and inconsistencies between 18 related studies evaluating XAI methods. To compare their results, colors coincide with the aggregated evaluation result of each XAI method across all metrics used in the study and belonging to one criterion. $n_A$: Amount of distinct and total XAI methods. $n_E$: Number of evaluation metrics. If $n_E = 0$, the study is conducted qualitatively.

datasets (as shown by Wang et al. [64], Rosenfeld [53]), a wide array of metrics have been introduced to evaluate XAI methods based on these three criteria quantitatively. These metrics are deployed in several studies (see Table 1) to determine *"What XAI method should I (not) use for my problem?"*. However, the current approach to quantitatively validate XAI methods has two major shortcomings, which we address in this work:

**Shortcoming 1: Gaps and inconsistencies in XAI evaluation.** Many studies restrict their analyses to a limited set of design parameters such as input modalities, (toy-)datasets, model architectures, attention or attribution-based XAI methods, and metrics, which all directly impact the performance of XAI methods (we define the first three parameters as *underlying* design parameters, as they directly influence the XAI method). Table 1 demonstrates this fragmented landscape specifically for the domain of computer vision, including discrepancies found across studies, with some methods, such as GradCAM (GC) [56], receiving contradictory assessments depending on the evaluation setup. As a consequence, our current understanding of XAI performance is limited, making it challenging for practitioners to determine a reliable XAI method for their specific use case.

**Shortcoming 2: Individual XAI metrics lack trustworthiness.** Recently, numerous metrics have been proposed to approximate the three stated XAI evaluation criteria. These metrics reflect the diversity of perspectives on the criterion. Studies typically apply one or two metrics to assess a certain criterion (see $n_E$ in Table 1). Arguably, this is not a reliable measure of success, as these limited subsets of perspectives on a criterion can lead to selection bias and overfitting to one metric or perspective. This selection bias becomes even more severe when reporting the mean across several metrics, which is a common procedure in XAI evaluation to summarize metric results [38, 29]. The small number of considered metrics along with their brittle and nontransparent aggregation diminishes the trustworthiness of results from recent XAI evaluation studies.

In response to these shortcomings, we present LATEC: the first comprehensive benchmark tailored for large-scale attribution & attention evaluation in computer vision. LATEC encompasses 17 of the most widely-used saliency XAI methods, including attention-based methods, and evaluates them using 20 distinct metrics (see Figure 1). Notably, LATEC integrates a variety of model architectures, and, to extend the evaluation spectrum beyond traditional 2D images, we included 3D point cloud and volume data, adapting XAI methods and metrics as necessary to suit these modalities. In total,

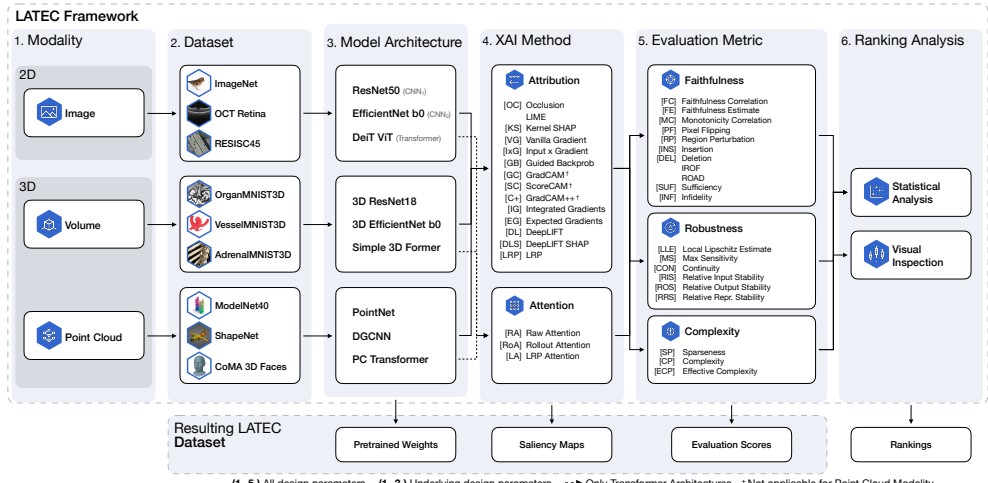

Figure 1: Structure of the LATEC framework including all design parameters and the output data of each stage provided as the LATEC dataset. Final rankings are analyzed in the benchmark.

LATEC assesses 7,560 unique combinations. LATEC addresses Shortcoming 1 by systematically incorporating all prevalent methods and metrics, as well as all vital underlying design parameters affecting XAI methods, and quantifying their effect on XAI methods. LATEC further addresses Shortcoming 2 by performing a dedicated analysis of the metrics themselves (also referred to as "meta-evaluation"), including a quantitative validation of the metrics' ranking behaviors, resulting in the identification of a more robust evaluation scheme. Moreover, in support of future research, we've made all intermediate data, including 326,790 saliency maps and 378,000 evaluation scores, as well as the benchmark publicly accessible.

## 2   The LATEC benchmark

The LATEC benchmark includes a framework and a dataset with the method rankings as the final output. The framework allows for diverse large-scale studies, structuring the experiments in six stages (see Figure 1), and the LATEC dataset provides reference data for evaluation and exploration. As the benchmark is easily extendable and leverages the high-quality dataset for standardized evaluation, it also serves as a foundation for future benchmarking of new XAI methods and metrics (see Appendix B for more information about the LATEC dataset).

**Utilized input datasets**    For the image modality, we use ImageNet (IMN) [19], UCSD OCT retina (OCT) [32] and RESISC45 (R45) [16], the volume modality the Adrenal-(AMN), Organ-(OMN) and VesselMedMNIST3D (VMN) datasets [69], and the point cloud modality the CoMA (CMA) [49], ModelNet40 (M40) [68] and ShapeNet (SHN) [13] datasets.

**Model architectures**    On each utilized dataset except IMN, where we take pretrained models, we train three models to achieve the architecture-dependent SOTA performance on the designated test set (if available, see Appendix A for a detailed description of the model training and hyperparameters). For the image modality, we use the ResNet50, EfficientNetb0, and DeiT ViT [63] architectures, for the volume modality the 3D ResNet18, 3D EfficientNetb0, and Simple3DFormer [66] architectures, and for the point cloud modality the PointNet, DGCNN and PC Transformer [26] architectures. The first two architectures are always CNNs and the third is a Transformer.

**XAI methods**    In total, we include 17 XAI methods, 14 attribution methods: Occlusion (OC) [72], LIME (on feature masks) [50], Kernel SHAP (KS, on feature masks) [39], Vanilla Gradient (VG) [58], Input x Gradient (IxG) [57], Guided Backprob (GB) [61], GC, ScoreCAM (SC) [65], GradCAM++ (C+) [14], Integrated Gradients (IG) [62], Expected Gradients (EG, also called Gradient SHAP) [23], DeepLIFT (DL) [57], DeepLIFT SHAP (DLS) [39], LRP (with $\epsilon$-,$\gamma$- and $0^+$-rules depending on the model architecture) [11], and three attention methods: Raw Attention (RA) [22], Rollout Attention (RoA) [1] and LRP Attention (LA) [15]. While the attribution methods are applied to all model architectures, the attention methods can only be applied to the Transformer-based architectures.

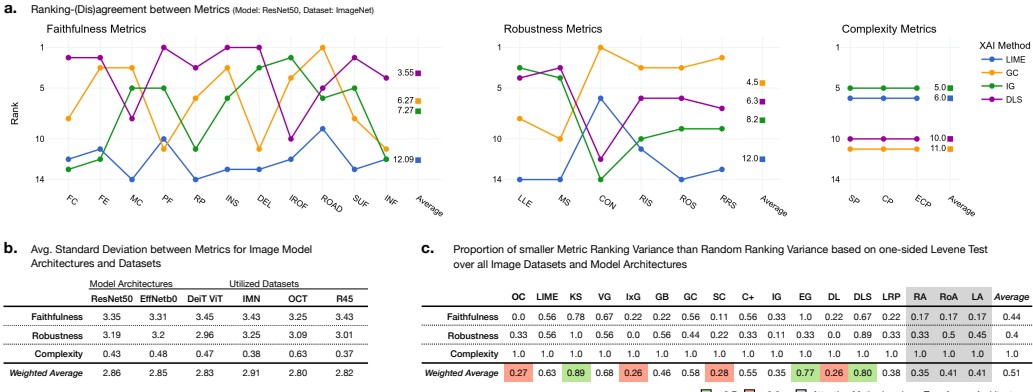

**a.** Ranking-(Dis)agreement between Metrics (Model: ResNet50, Dataset: ImageNet)

**b.** Avg. Standard Deviation between Metrics for Image Model Architectures and Datasets

| | Model Architectures | | | Utilized Datasets | | |
|---|---|---|---|---|---|---|
| | ResNet50 | EffNetb0 | DeiT ViT | IMN | OCT | R45 |
| Faithfulness | 3.35 | 3.31 | 3.45 | 3.43 | 3.25 | 3.43 |
| Robustness | 3.19 | 3.2 | 2.96 | 3.25 | 3.09 | 3.01 |
| Complexity | 0.43 | 0.48 | 0.47 | 0.38 | 0.63 | 0.37 |
| *Weighted Average* | 2.86 | 2.85 | 2.83 | 2.91 | 2.80 | 2.82 |

**c.** Proportion of smaller Metric Ranking Variance than Random Ranking Variance based on one-sided Levene Test over all Image Datasets and Model Architectures

| | OC | LIME | KS | VG | IxG | GB | GC | SC | C+ | IG | EG | DL | DLS | LRP | RA | RoA | LA | *Average* |
|---|---|---|---|---|---|---|---|---|---|---|---|---|---|---|---|---|---|---|
| Faithfulness | 0.0 | 0.56 | 0.78 | 0.67 | 0.22 | 0.22 | 0.56 | 0.11 | 0.56 | 0.33 | 1.0 | 0.22 | 0.67 | 0.22 | 0.17 | 0.17 | 0.17 | 0.44 |
| Robustness | 0.33 | 0.56 | 1.0 | 0.56 | 0.0 | 0.56 | 0.44 | 0.22 | 0.33 | 0.11 | 0.33 | 0.0 | 0.89 | 0.33 | 0.33 | 0.5 | 0.45 | 0.4 |
| Complexity | 1.0 | 1.0 | 1.0 | 1.0 | 1.0 | 1.0 | 1.0 | 1.0 | 1.0 | 1.0 | 1.0 | 1.0 | 1.0 | 1.0 | 1.0 | 1.0 | 1.0 | 1.0 |
| *Weighted Average* | 0.27 | 0.63 | 0.89 | 0.68 | 0.26 | 0.46 | 0.58 | 0.28 | 0.55 | 0.35 | 0.77 | 0.26 | 0.80 | 0.38 | 0.35 | 0.41 | 0.41 | 0.51 |

■ > 0.7  ■ < 0.3  ▨ Attention Methods only on Transformer Architectures

Figure 2: **a.** Ranking of four XAI methods based on all evaluation metrics of each criterion for one specific set of design parameters. **b.** Average standard deviation per model architectures and utilized datasets for the imaging modality. The weighted average per column is based on the number of metrics per criterion. **c.** Proportion of accepted one-sided Levene-Tests for significantly smaller ranking variance compared to the variance of an entire random ranking. Larger values show higher agreement between metrics. The weighted average is based on the number of metrics per criterion.

Related work by Hooker et al. [30] and Yang and Kim [70] showed that advancing methods by VarGrad [2] or SmoothGrad [60] can, in general, improve results. We conduct an ablation study in Appendix O to validate these findings for our benchmark. Contrary to Hooker et al. [30], we find no substantial improvements w.r.t faithfulness or robustness, only SmoothGrad notably reduces complexity by producing more localized saliency maps. Thus, we only consider the original methods without adaptations in the benchmark.

We qualitatively tuned the XAI hyperparameters per dataset (see Appendix B), as also commonly done to avoid biasing the quantitative evaluation results (see Appendix C for all hyperparameters). We observe that most hyperparameters generalize well across the datasets within a modality. To further validate the non-sensitivity of the benchmark rankings to reasonably selected hyperparameters, we conduct an ablation study including the top five ranked XAI methods in Appendix N, validating the robustness of their performance.

**Evaluation metrics** We utilize a total of 20 well-established evaluation metrics, which are grouped into three criteria: faithfulness ("Is the explanation following the model behavior?"), robustness ("Is the explanation stable?"), and complexity ("Is the explanation concise and understandable?"). 11 metrics evaluate faithfulness: Faithfulness Correlation (FC) [9], Faithfulness Estimate (FE) [44], Pixel Flipping (PF) [8], Region Perturbation (RP) [55], Insertion (INS) and Deletion (DEL) [47], Iterative Removal of Features (IROF) [51], Remove and Debias (ROAD) [52], Sufficiency (SUF) [18] and Infidelity (INF) [71], 6 metrics evaluate robustness: Local Lipschitz Estimate (LLE) [5], Max Sensitivity (MS) [71], Continuity (CON) [42] and Relative Input/Output/Representation Stability (RIS, ROS, RRS) [4], and 3 metrics evaluate complexity: Sparseness (SP) [12], Complexity (CP) and Effective Complexity (ECP) [44]. See Appendix D for a description of every metric. We select the hyperparameters per dataset as some depend on dataset properties (see Appendix subsection D.2 for all parameters). As the raw evaluation scores have no semantic meaning and can be extremely skewed in their distributions, making them hard to interpret and compare, we analyze the XAI methods and metrics based on their ranking.

**3D adaptation** While several XAI methods and metrics in LATEC are independent of the input space dimensions, others had to be adapted to 3D volume and point cloud data, building upon the implementations for image data by Kokhlikyan et al. [34] and Hedström et al. [28]. Due to the adaptations, this benchmark provides the first large-scale insights into XAI method performance and metric behavior on 3D data. We describe the adaptation process, including rigorous testing, for all respective XAI methods and metrics in Appendix H and show illustrative saliency maps.

# 3  Deriving a reliable evaluation scheme for XAI

## 3.1  How severe is the risk of metric selection bias in XAI evaluation?

In Shortcoming 2, we describe a risk of selection bias due to approximating an evaluation criterion with only one or a few metrics, possibly overfitting to a limited set of perspectives on the criterion. In this metrics analysis, we first aim to provide empirical evidence for this risk. A first exploratory analysis quickly supports the hypothesis, as we encounter strong ranking disagreement between metrics for various combinations of underlying design parameters. We define ranking agreement as the consensus among metrics belonging to one criterion about the rank of one XAI method when evaluating and subsequently ranking this method against all other XAI methods. Consequently, disagreement in ranking is defined through high variance between the determined ranks of the metrics for one XAI method. For example, Figure 2 (a.) demonstrates the ranking behavior of four selected XAI methods for one selection of underlying design parameters. The line charts show how each metric ranks the four XAI methods in comparison to all other XAI methods, with the mean aggregated average rank to the right. For faithfulness, we observe high disagreement between metrics in their ranking of GC and IG, mainly agreeing metrics in the ranking of DLS (with IROF and MC being noticeable outliers), and agreeing metrics in the case of LIME.

**Do metric disagreements depend on underlying design parameters?** The inquiry emerges as to whether the risk of selection bias is generally present in certain combinations of underlying design parameters or is uniformly distributed across them. To this end, we computed the average standard deviation (SD) between metric rankings either aggregated across model architectures or datasets (see Appendix Equation 1 and Equation 2 for the mathematical formulation) to observe the variance among metrics per modality, model architecture, and datasets. Figure 2 (b.) shows for the imaging modality that the average standard deviation is generally stable between model architectures or datasets within each evaluation criterion (see Appendix I for the other two modalities). Thus, we can conclude that there is no single model architecture, modality, or dataset choice that has a substantial effect on the disagreement between metric rankings of all XAI methods.

**Do metric disagreements vary for individual XAI methods?** Now that we can rule out the general influence of underlying design parameters, we quantify how strong the risk of metric disagreement is in general and if there is a difference between XAI methods. To this end, we utilize a one-sided Levene's Test [37], testing if the rank-variance of a set of metrics is significantly lower than the variance of a random rank distribution, which can be analytically inferred. We compute this test for all sets of metrics on every possible combination of design parameters and mean aggregate over model architectures and datasets (see Appendix Equation 3 for the mathematical formulation). Figure 2 (c.) shows for the imaging modality the resulting proportion of accepted tests ($\alpha = 0.1$) for each criterion and XAI method. By computing the weighted average proportion across criteria at the bottom, we indeed observe strong variations between the XAI methods. Specifically for KS, EG, and DLS, in a large majority of cases, metrics agree, while for OC, IxG, SC, and DL only in about $\sim 27\%$ of the cases variance in metric ranking is significantly lower than random ranking. Concluding, our findings reveal that metrics disagree and agree in varying degrees depending on the XAI method. We refer to Appendix J for a study on why metrics disagree.

## 3.2  How can we identify reliable trends within agreeing and disagreeing metrics?

After providing empirical evidence for the risk of selection bias in XAI evaluation, we identify three resulting major limitations **(1-3)** in current practice, collectively summarized as Shortcoming 2. The current practice in XAI evaluation is to employ a small set of metrics and subsequently mean aggregate over the normalized scores to increase the generalization of results and simplify data analysis in big datasets (see e.g. Li et al. [38], Hedström et al. [28], Hesse et al. [29]). We present each limitation of this procedure with a corresponding solution and combine them into a single evaluation scheme. This proposed scheme aims to reliably benchmark XAI methods across both agreeing and disagreeing metrics, offering a more robust evaluation approach.

**(1)** Current practice includes only small sets of metrics, which comes with an increased risk of biases due to a high dependence on metric selection and individual metric behavior. Further, mean aggregating in such small samples lacks robustness against "outlier metrics". The subsection 3.1 demonstrates this risk of selection bias extensively. We start addressing this limitation by including the to-date largest scale of diverse and relevant metrics, minimizing the risk of selection bias and

the influence of outliers metrics. The adequacy of the selected metrics is ensured by exclusively choosing standard metrics commonly implemented in widely used software libraries [27, 34], while also ensuring diversity in their implementation and interpretation of the respective criteria. This approach is crucial to avoid overfitting to a single perspective.

(**2**) Aggregation across metrics obfuscates their diverse perspective on the approximated evaluation criterion. Such aggregation can be further flawed due to unbounded metrics, inconsistent interpretations (e.g. correlation coefficient vs. distance-based metrics), and sensitivity to metric score outliers and distribution skewness (also shown by Colombo et al. [17]). We address this limitation by employing an "aggregate-then-rank" scheme, which is already well established for large-scale evaluation and benchmarks of model performance [40, 17, 54]. By aggregating the median evaluation score of all model and dataset combinations of one modality and metric we get more robust metric scores without ignoring the perspective of the individual metric. Subsequently, we rank the computed scores, as rankings are independent of the scales or units of the metrics and are generally easier to interpret [54]. However, we acknowledge that in a large-scale study such as ours, abstraction is necessary to distill meaningful results from the extensive set of rankings. To highlight strong trends across metrics we compute their average rank per XAI method, indicated by "$\hat{\mu}$" in Table 2. A consistent high or low average rank for an XAI method implies a general agreement among the metrics evaluating a specific criterion.

(**3**) Current studies ignore the extent of disagreement between metrics, which contains crucial information about a method's performance. We believe that understanding why metrics disagree and the situations in which this occurs is vital for evaluating XAI. To determine the presence of disagreement in general, we deployed the Levene test in Figure 2. In the XAI benchmark, however, we are interested in comparing the level of disagreement between XAI methods. To this end, we calculate the SD between ranks of an XAI method as a measure of disagreement, indicated as "$\hat{\sigma}$".

**Proposed evaluation scheme.** We subsequently combine all solutions into one proposed evaluation scheme. To include all metric perspectives and increase general ranking robustness, we first calculate the median of the standardized evaluation score from all our included relevant metrics for each combination of dataset and model. Further, we average these medians and rank the methods according to each metric (see Appendix F for a detailed flow chart of how we get from evaluation scores to rankings). Ranking XAI methods across one metric's evaluation scores from several input datasets and model architectures makes the ranking more robust to variation within these parameters. To analyze the large set of ranking results, we jointly utilize the mean $\hat{\mu}$, median $x_{n/2}$, and SD $\hat{\sigma}$ (e.g. in Table 2) to detect strong ranking trends through $\hat{\mu}$ and $x_{n/2}$, and determine their trustworthiness through $\hat{\sigma}$, before focusing onto the individual metrics. We determine the threshold values in $\hat{\sigma}$, indicating high or low SD, based on the quantiles of each evaluation criterion's SD distribution. To assess the statistical significance of the differences between two methods, we report the p-values of the Wilcoxon-Mann-Whitney tests for all modalities and evaluation criteria, comparing the rankings of all XAI methods, as detailed in Appendix P. Based on this evaluation scheme, we achieve more robust rankings, include a diverse set of metric perspectives, and still leverage a mechanism to highlight strong trends and (dis-)agreeing metrics from the large set of results.

### 3.3 Additional insights for robust evaluation

We encountered further pitfalls of the current metric application in XAI, which to our knowledge have not been discussed before. While all pitfalls are discussed in Appendix L, one pitfall regarding complexity evaluation is in our opinion especially critical. Suspiciously, Figure 2 (c.) indicates almost no disagreement between complexity rankings. Further, CAM (GC, SC, C+) and attention (RA, RoA, LA) methods are ranked significantly more complex (see Table 2). In our opinion, this observation is counter-intuitive when comparing the complexity rankings to the saliency maps in Appendix B, based on which we would classify CAM and attention methods as more localized and less noisy. While all three complexity metrics are also explicitly proposed for image data, we notice that they all treat each pixel, voxel, or point independently of each other, ignoring locality and favoring methods that attribute to the smallest set of single pixels. As this approach possibly transfers to low dimensional images such as MNIST [36] or CIFAR-10 [35], the image datasets the three metrics are originally presented on, we hypothesize that it may not be effective with higher-dimensional inputs as observed in our study. Consequently, it is expected that techniques such as LRP would be highly regarded due to their emphasis on filtering the significance of individual pixels, in contrast to CAM methods

**a. Image Modality**

Faithfulness

| | $\hat{\mu}$ | $x_{n/2}$ | $\hat{\sigma}$ | FC | FE | MC | PF | RP | INS | DEL | IROF | ROAD | SUF | INF |
|---|---|---|---|---|---|---|---|---|---|---|---|---|---|---|
| OC | 9.3 | 10.0 | 5.1 | 2 | 3 | 14 | 16 | 3 | 12 | 15 | 5 | 8 | 10 | 14 |
| LIME | 14.7 | 16.0 | 2.9 | 14 | 17 | 17 | 17 | 16 | 17 | 17 | 12 | 10 | 9 | 16 |
| KS | 14.3 | 15.0 | 2.7 | 13 | 16 | 15 | 15 | 17 | 16 | 16 | 13 | 11 | 8 | 17 |
| VG | 10.7 | 11.0 | 3.0 | 11 | 11 | 6 | 9 | 14 | 15 | 9 | 14 | 13 | 6 | 10 |
| IxG | 7.1 | 7.0 | 3.9 | 3 | 14 | 3 | 7 | 7 | 8 | 5 | 8 | 15 | 4 | 4 |
| GB | 7.6 | 8.0 | 5.3 | 1 | 1 | 16 | 2 | 9 | 10 | 8 | 10 | 17 | 7 | 3 |
| GC | 8.0 | 9.0 | 3.3 | 9 | 7 | 4 | 11 | 10 | 11 | 12 | 3 | 4 | 12 | 5 |
| SC | 7.9 | 9.0 | 4.9 | 8 | 9 | 12 | 12 | 2 | 9 | 13 | 1 | 3 | 16 | 2 |
| C+ | 9.5 | 10.0 | 3.7 | 10 | 10 | 11 | 13 | 12 | 13 | 14 | 2 | 5 | 5 | 9 |
| IG | 7.5 | 6.0 | 3.8 | 4 | 13 | 5 | 10 | 6 | 5 | 1 | 6 | 12 | 13 | 8 |
| EG | 5.9 | 6.0 | 3.4 | 6 | 2 | 2 | 4 | 8 | 1 | 4 | 9 | 7 | 11 | 11 |
| DL | 7.5 | 7.0 | 3.4 | 7 | 12 | 7 | 6 | 11 | 6 | 2 | 7 | 14 | 3 | 7 |
| DLS | 4.8 | 4.0 | 3.1 | 5 | 6 | 1 | 5 | 4 | 2 | 3 | 4 | 9 | 2 | 12 |
| LRP | 12.5 | 14.0 | 4.0 | 12 | 15 | 13 | 14 | 15 | 14 | 11 | 11 | 16 | 1 | 15 |
| RA | 9.4 | 8.0 | 4.7 | 16 | 5 | 4 | 8 | 13 | 7 | 10 | 17 | 6 | 14 | 1 |
| RoA | 8.8 | 8.0 | 5.2 | 17 | 8 | 9 | 3 | 5 | 3 | 7 | 15 | 2 | 15 | 13 |
| LA | 7.5 | 6.0 | 6.0 | 16 | 5 | 10 | 1 | 1 | 4 | 6 | 16 | 1 | 17 | 6 |

Robustness

| | $\hat{\mu}$ | $x_{n/2}$ | $\hat{\sigma}$ | LLE | MS | CON | RIS | ROS | RRS |
|---|---|---|---|---|---|---|---|---|---|
| OC | 9.0 | 8.0 | 4.4 | 15 | 14 | 2 | 8 | 7 | 8 |
| LIME | 14.5 | 16.5 | 4.7 | 16 | 16 | 4 | 17 | 17 | 17 |
| KS | 14.5 | 16.0 | 4.3 | 17 | 17 | 5 | 16 | 16 | 16 |
| VG | 5.5 | 5.0 | 2.6 | 7 | 8 | 9 | 3 | 3 | 3 |
| IxG | 9.8 | 11.0 | 4.3 | 5 | 4 | 8 | 14 | 14 | 14 |
| GB | 8.0 | 9.0 | 3.7 | 2 | 5 | 7 | 12 | 11 | 11 |
| GC | 6.5 | 6.0 | 3.3 | 8 | 12 | 1 | 6 | 6 | 6 |
| SC | 10.5 | 9.5 | 3.6 | 11 | 13 | 17 | 7 | 8 | 7 |
| C+ | 12.3 | 12.0 | 2.7 | 14 | 15 | 16 | 10 | 10 | 9 |
| IG | 8.3 | 11.5 | 5.2 | 1 | 1 | 13 | 11 | 12 | 12 |
| EG | 4.2 | 3.5 | 3.3 | 9 | 7 | 6 | 1 | 1 | 1 |
| DL | 9.3 | 12.5 | 4.9 | 3 | 2 | 12 | 13 | 13 | 13 |
| DLS | 7.7 | 9.0 | 3.0 | 4 | 3 | 11 | 9 | 9 | 10 |
| LRP | 11.2 | 12.5 | 4.1 | 6 | 6 | 10 | 15 | 15 | 15 |
| RA | 6.5 | 5.5 | 4.8 | 10 | 9 | 14 | 2 | 2 | 2 |
| RoA | 7.0 | 5.0 | 3.7 | 13 | 11 | 3 | 5 | 5 | 5 |
| LA | 8.2 | 7.0 | 4.4 | 12 | 10 | 15 | 4 | 4 | 4 |

Complexity

| | $\hat{\mu}$ | $x_{n/2}$ | $\hat{\sigma}$ | SP | CP | ECP |
|---|---|---|---|---|---|---|
| OC | 10.0 | 10.0 | 0.8 | 10 | 9 | 11 |
| LIME | 6.7 | 6.0 | 0.9 | 6 | 6 | 8 |
| KS | 14.3 | 15.0 | 1.7 | 16 | 15 | 12 |
| VG | 12.3 | 13.0 | 1.7 | 13 | 14 | 10 |
| IxG | 4.3 | 4.0 | 0.5 | 4 | 5 | 4 |
| GB | 4.7 | 5.0 | 0.5 | 5 | 4 | 5 |
| GC | 12.0 | 12.0 | 0.8 | 12 | 11 | 13 |
| SC | 8.0 | 8.0 | 0.8 | 9 | 8 | 7 |
| C+ | 14.0 | 14.0 | 0.8 | 14 | 13 | 15 |
| IG | 3.0 | 3.0 | 0.0 | 3 | 3 | 3 |
| EG | 17.0 | 17.0 | 0.0 | 17 | 17 | 17 |
| DL | 2.0 | 2.0 | 0.0 | 2 | 2 | 2 |
| DLS | 7.0 | 7.0 | 0.8 | 8 | 7 | 6 |
| LRP | 1.0 | 1.0 | 0.0 | 1 | 1 | 1 |
| RA | 8.7 | 9.0 | 1.2 | 7 | 10 | 9 |
| RoA | 15.7 | 16.0 | 0.5 | 15 | 16 | 16 |
| LA | 12.3 | 12.0 | 1.2 | 11 | 12 | 14 |

$x_{n/2}, \hat{\mu}$ : □ < 7  □ > 10      $\hat{\sigma}$ : □ < 0.15 Quantile  □ > 0.85 Quantile

**b. Volume Modality**

Faithfulness

| | $\hat{\mu}$ | $x_{n/2}$ | $\hat{\sigma}$ | FC | FE | MC | PF | RP | INS | DEL | IROF | ROAD | SUF | INF |
|---|---|---|---|---|---|---|---|---|---|---|---|---|---|---|
| OC | 10.6 | 10.0 | 4.7 | 17 | 17 | 17 | 10 | 14 | 4 | 10 | 4 | 8 | 7 | 9 |
| LIME | 8.8 | 9.0 | 4.8 | 3 | 3 | 13 | 17 | 6 | 13 | 5 | 13 | 3 | 3 | 12 |
| KS | 8.8 | 8.0 | 5.0 | 4 | 4 | 4 | 14 | 16 | 8 | 11 | 11 | 3 | 17 | 5 |
| VG | 10.2 | 10.0 | 4.5 | 5 | 5 | 5 | 15 | 6 | 10 | 15 | 16 | 7 | 15 | 13 |
| IxG | 5.8 | 6.0 | 3.7 | 6 | 6 | 6 | 4 | 10 | 15 | 3 | 7 | 2 | 2 | 3 |
| GB | 6.7 | 7.0 | 3.5 | 7 | 7 | 7 | 6 | 8 | 14 | 7 | 10 | 1 | 6 | 1 |
| GC | 12.8 | 14.0 | 3.5 | 8 | 8 | 8 | 17 | 13 | 16 | 16 | 14 | 15 | 16 | 10 |
| SC | 8.4 | 11.0 | 4.6 | 2 | 2 | 2 | 11 | 11 | 5 | 12 | 8 | 11 | 14 | 14 |
| C+ | 11.5 | 9.0 | 3.8 | 9 | 9 | 9 | 16 | 9 | 12 | 17 | 17 | 14 | 8 | 6 |
| IG | 8.6 | 11.0 | 4.3 | 12 | 12 | 12 | 3 | 5 | 13 | 4 | 13 | 9 | 1 | 11 |
| EG | 7.0 | 7.0 | 3.9 | 11 | 11 | 11 | 1 | 7 | 9 | 1 | 6 | 4 | 2 | 4 |
| DL | 7.2 | 6.0 | 4.5 | 13 | 13 | 13 | 2 | 12 | 7 | 2 | 5 | 6 | 4 | 2 |
| DLS | 9.4 | 10.0 | 3.9 | 14 | 14 | 14 | 9 | 4 | 11 | 5 | 3 | 12 | 3 | 7 |
| LRP | 13.0 | 15.0 | 3.4 | 15 | 15 | 15 | 12 | 15 | 17 | 14 | 15 | 12 | 5 | 8 |
| RA | 10.1 | 11.0 | 5.9 | 16 | 16 | 16 | 8 | 2 | 1 | 9 | 2 | 13 | 12 | 17 |
| RoA | 7.9 | 10.0 | 5.2 | 10 | 10 | 10 | 5 | 1 | 2 | 6 | 1 | 17 | 10 | 15 |
| LA | 6.2 | 3.0 | 5.4 | 1 | 1 | 1 | 7 | 3 | 3 | 8 | 3 | 16 | 9 | 16 |

Robustness

| | $\hat{\mu}$ | $x_{n/2}$ | $\hat{\sigma}$ | LLE | MS | CON | RIS | ROS | RRS |
|---|---|---|---|---|---|---|---|---|---|
| OC | 9.7 | 11.0 | 4.0 | 12 | 12 | 1 | 13 | 10 | 10 |
| LIME | 12.3 | 15.0 | 5.1 | 16 | 16 | 2 | 10 | 15 | 15 |
| KS | 14.5 | 16.0 | 4.3 | 17 | 17 | 5 | 16 | 16 | 16 |
| VG | 5.7 | 5.5 | 4.7 | 10 | 10 | 11 | 1 | 1 | 1 |
| IxG | 10.8 | 13.5 | 4.5 | 5 | 4 | 15 | 14 | 14 | 13 |
| GB | 10.8 | 11.5 | 3.1 | 7 | 7 | 16 | 11 | 12 | 12 |
| GC | 9.8 | 8.5 | 2.7 | 14 | 13 | 8 | 8 | 7 | 9 |
| SC | 10.2 | 8.5 | 3.5 | 15 | 15 | 9 | 7 | 8 | 7 |
| C+ | 10.3 | 9.5 | 4.5 | 13 | 14 | 17 | 6 | 6 | 6 |
| IG | 6.8 | 8.5 | 4.0 | 2 | 1 | 12 | 9 | 9 | 9 |
| EG | 5.3 | 4.5 | 3.8 | 6 | 6 | 13 | 2 | 2 | 3 |
| DL | 8.2 | 9.0 | 4.1 | 3 | 3 | 7 | 12 | 13 | 11 |
| DLS | 9.8 | 10.5 | 4.1 | 4 | 5 | 10 | 15 | 11 | 14 |
| LRP | 11.3 | 15.5 | 7.0 | 1 | 2 | 14 | 17 | 17 | 17 |
| RA | 6.0 | 5.5 | 2.4 | 9 | 9 | 6 | 5 | 5 | 2 |
| RoA | 5.0 | 4.0 | 2.2 | 8 | 8 | 3 | 4 | 3 | 4 |
| LA | 6.3 | 4.5 | 3.3 | 11 | 11 | 4 | 3 | 4 | 5 |

Complexity

| | $\hat{\mu}$ | $x_{n/2}$ | $\hat{\sigma}$ | SP | CP | ECP |
|---|---|---|---|---|---|---|
| OC | 4.7 | 6.0 | 2.6 | 7 | 1 | 6 |
| LIME | 11.7 | 11.0 | 2.5 | 9 | 15 | 11 |
| KS | 11.7 | 12.0 | 2.1 | 12 | 14 | 9 |
| VG | 14.3 | 15.0 | 0.9 | 15 | 13 | 15 |
| IxG | 5.3 | 2.0 | 4.7 | 2 | 12 | 2 |
| GB | 8.3 | 8.0 | 2.1 | 6 | 11 | 8 |
| GC | 11.3 | 10.0 | 1.9 | 14 | 10 | 10 |
| SC | 15.3 | 16.0 | 1.7 | 17 | 16 | 13 |
| C+ | 13.0 | 14.0 | 2.9 | 16 | 9 | 14 |
| IG | 4.7 | 4.0 | 1.7 | 4 | 7 | 3 |
| EG | 7.0 | 7.0 | 0.8 | 8 | 6 | 7 |
| DL | 4.0 | 4.0 | 0.8 | 3 | 5 | 4 |
| DLS | 4.7 | 5.0 | 0.5 | 5 | 4 | 5 |
| LRP | 1.7 | 1.0 | 0.9 | 1 | 3 | 1 |
| RA | 8.3 | 11.0 | 4.5 | 11 | 2 | 12 |
| RoA | 11.3 | 10.0 | 3.4 | 10 | 8 | 16 |
| LA | 15.7 | 17.0 | 1.9 | 13 | 17 | 17 |

$x_{n/2}, \hat{\mu}$ : □ < 7  □ > 10      $\hat{\sigma}$ : □ < 0.15 Quantile  □ > 0.85 Quantile

**c. Point Cloud Modality**

Faithfulness

| | $\hat{\mu}$ | $x_{n/2}$ | $\hat{\sigma}$ | FC | FE | MC | PF | RP | INS | DEL | IROF | ROAD | SUF | INF |
|---|---|---|---|---|---|---|---|---|---|---|---|---|---|---|
| OC | 8.5 | 10.0 | 4.3 | 1 | 14 | 13 | 11 | 6 | 7 | 10 | 10 | 14 | 4 | 3 |
| LIME | 5.2 | 6.0 | 2.7 | 8 | 1 | 1 | 6 | 8 | 1 | 6 | 6 | 6 | 6 | 8 |
| KS | 7.0 | 7.0 | 3.6 | 6 | 7 | 8 | 2 | 1 | 5 | 8 | 11 | 10 | 5 | 14 |
| VG | 7.9 | 9.0 | 3.4 | 9 | 9 | 9 | 8 | 2 | 10 | 11 | 1 | 7 | 8 | 13 |
| IxG | 6.7 | 5.0 | 4.0 | 7 | 4 | 2 | 4 | 3 | 9 | 5 | 5 | 11 | 14 | 12 |
| GB | 5.3 | 4.0 | 3.8 | 4 | 2 | 3 | 1 | 10 | 8 | 1 | 2 | 5 | 11 | 11 |
| IG | 7.5 | 7.0 | 3.1 | 10 | 6 | 5 | 5 | 7 | 4 | 3 | 8 | 9 | 12 | 9 |
| EG | 8.0 | 9.0 | 3.2 | 11 | 13 | 10 | 9 | 5 | 3 | 2 | 7 | 9 | 9 | 10 |
| DL | 5.8 | 4.0 | 3.8 | 14 | 5 | 4 | 3 | 4 | 2 | 4 | 4 | 8 | 13 | 6 |
| DLS | 8.2 | 8.0 | 2.2 | 12 | 11 | 7 | 7 | 9 | 6 | 9 | 4 | 8 | 10 | 7 |
| LRP | 9.4 | 10.0 | 2.6 | 13 | 12 | 6 | 10 | 11 | 7 | 7 | 9 | 12 | 7 | 5 |
| RA | 8.0 | 11.0 | 5.4 | 2 | 3 | 11 | 14 | 12 | 14 | 12 | 3 | 2 | 1 | |
| RoA | 8.7 | 12.0 | 4.5 | 5 | 8 | 12 | 12 | 12 | 13 | 12 | 14 | 1 | 3 | 4 |
| LA | 8.9 | 13.0 | 5.3 | 10 | 14 | 13 | 13 | 14 | 13 | 13 | 2 | 1 | 2 | |

Robustness

| | $\hat{\mu}$ | $x_{n/2}$ | $\hat{\sigma}$ | LLE | MS | CON | RIS | ROS | RRS |
|---|---|---|---|---|---|---|---|---|---|
| OC | 10.5 | 10.0 | 1.9 | 13 | 13 | 10 | 8 | 10 | 9 |
| LIME | 13.2 | 14.0 | 1.9 | 14 | 14 | 9 | 14 | 14 | 14 |
| KS | 9.0 | 8.5 | 2.3 | 12 | 12 | 6 | 7 | 9 | 8 |
| VG | 6.7 | 5.0 | 2.4 | 10 | 10 | 5 | 5 | 5 | 5 |
| IxG | 8.8 | 9.0 | 3.3 | 7 | 6 | 4 | 12 | 11 | 13 |
| GB | 8.8 | 9.0 | 1.1 | 9 | 9 | 7 | 10 | 8 | 10 |
| IG | 7.3 | 8.0 | 4.2 | 5 | 4 | 1 | 11 | 12 | 11 |
| EG | 7.5 | 7.5 | 3.5 | 11 | 11 | 11 | 4 | 4 | 4 |
| DL | 9.0 | 9.5 | 3.9 | 6 | 7 | 3 | 13 | 13 | 12 |
| DLS | 7.0 | 7.0 | 1.0 | 8 | 8 | 8 | 6 | 6 | 6 |
| LRP | 5.7 | 6.0 | 2.3 | 4 | 5 | 2 | 9 | 7 | 7 |
| RA | 4.7 | 3.0 | 4.2 | 3 | 3 | 14 | 3 | 2 | 3 |
| RoA | 4.0 | 2.0 | 4.0 | 2 | 2 | 13 | 2 | 3 | 2 |
| LA | 2.8 | 1.0 | 4.1 | 1 | 1 | 12 | 1 | 1 | 1 |

Complexity

| | $\hat{\mu}$ | $x_{n/2}$ | $\hat{\sigma}$ | SP | CP | ECP |
|---|---|---|---|---|---|---|
| OC | 8.3 | 8.0 | 2.1 | 11 | 8 | 6 |
| LIME | 8.3 | 8.0 | 1.2 | 8 | 10 | 7 |
| KS | 9.3 | 9.0 | 0.5 | 10 | 9 | 9 |
| VG | 8.0 | 7.0 | 1.4 | 7 | 7 | 10 |
| IxG | 2.0 | 2.0 | 0.8 | 3 | 2 | 1 |
| GB | 4.0 | 4.0 | 0.0 | 4 | 4 | 4 |
| IG | 2.7 | 3.0 | 0.5 | 2 | 3 | 3 |
| EG | 10.3 | 11.0 | 0.9 | 9 | 11 | 11 |
| DL | 1.3 | 1.0 | 0.5 | 1 | 1 | 2 |
| DLS | 6.3 | 6.0 | 1.2 | 5 | 6 | 8 |
| LRP | 5.3 | 5.0 | 0.5 | 6 | 5 | 5 |
| RA | 12.0 | 12.0 | 0.0 | 12 | 12 | 12 |
| RoA | 13.7 | 14.0 | 0.5 | 14 | 14 | 13 |
| LA | 13.3 | 13.0 | 0.5 | 13 | 13 | 14 |

$x_{n/2}, \hat{\mu}$ : □ < 6  □ > 9      $\hat{\sigma}$ : □ < 0.15 Quantile  □ > 0.85 Quantile

Table 2: Metric rankings of the XAI methods for all three modalities based on the ranking computation across model architectures and datasets outlined in section 2. For each XAI method $\hat{\mu}$ (mean) and $x_{n/2}$ (median) indicate strong ranking trends, and $\hat{\sigma}$ (SD) high or low disagreement between metrics. Coloring of $\hat{\mu}$ and $x_{n/2}$ coincide with top and bottom positions as point cloud rankings are of length 14 and all others are of length 17. $\hat{\sigma}$ coloring coincides with the upper 0.15 and red with the lower 0.85 quantiles of each evaluation criterion.

that attribute importance to broader local regions. Whether these XAI methods are less complex and more human-understandable on computer vision modalities is debatable and subsequent complexity evaluation results should be interpreted with caution.

## 4 XAI benchmark results

Based on the proposed evaluation scheme, we highlight six main findings, each of which consists of an *observation* followed by a *recommendation*. These findings encompass general trends across design parameters as well as specific trends pertinent to particular XAI methods. We ensure the robustness of highlighted results by focusing on observations with low SD between metrics.

**Expected Gradients consistently rank among the top methods in terms of faithfulness and robustness.** However, no XAI method ranks consistently high on all evaluation criteria and modalities. Further, EG consistently exhibits an average SD between metrics. We would recommend EG as an initial approach in various situations, also due to its low dependence on hyperparameter selection and input modalities, especially when baseline values are non-trivial to select.

**Rankings of XAI methods typically generalize well over datasets and model architectures.** The tables in Appendix K show minimal ranking disparities between datasets or model architectures within individual modalities. This suggests that a method selected for one dataset or model architecture

can transfer well to others if data dimensionality and characteristics are not too distinct. For model architectures, however, there are two notable exceptions. CAM methods generally show higher ranking dissimilarities between architectures, which could be attributed to differences in latent representations of the models, as the semantics captured in the last convolutional or cls-token layers do not have to coincide between models. Thus, we recommend increasing the robustness by averaging the activation map of several hidden layers, which has shown effective in application [25], but can lead to less localized saliency maps. LRP shows additionally high dissimilarity between CNN and Transformer architectures, especially for the 3D modalities. We use the recommended $\gamma$- and $\epsilon$-rules in LRP for the CNN models. However, on Transformer architectures, LRP does not preserve the conservation rule and only works with the $0^+$-rule (see Chefer et al. [15]). Both implemented changes to LRP bias the relevance computation, which consequentially impacts its performance on Transformer architectures. Thus, we recommend using LA instead of LRP as a relevance-based method on Transformer architectures, as it leverages the Transformer-inherent attention and performs better regarding faithfulness and robustness.

**Ranks of XAI methods are highly dependent on the input modality, especially for linear surrogate and CAM methods.** In general, we observe substantial ranking differences across modalities. Especially both linear surrogate methods (LIME, KS) underperform on image and volume compared to the lower dimensional point cloud modality in terms of faithfulness. On these modalities, their performance also strongly depends on the suitability of the feature mask computed via a grid or super-pixels, which is very time-consuming to fine-tune for single observations. Further, their evaluated robustness is very low across all modalities. Concluding, we advise against using them for high-dimensional and complex relationships. CAM methods achieve always higher faithfulness on image than on volume data. When comparing the saliency maps between both modalities, we observe that the volume-based maps are much coarser (i.e. more "blocky") and less focused. We attribute this observation to less accurate latent model representations and subsequent up-sampling in 3D compared to 2D space, subsequently not recommending them for volume data. Overall, results indicate higher consistency in robustness across modalities, with greater variability in faithfulness and complexity. Notably, the standard deviation of robustness metrics differs significantly between volume and point cloud data, suggesting an influence on metric disagreement.

**Attention methods are more robust compared to attribution methods but can exhibit strong disagreement among metrics** We observe a very large SD for the three attention methods (RA, RoA, LA) as faithfulness and robustness metrics rank them either very high or low (except for robustness on volume data). Therefore, we would strongly recommend investigating the interaction between metrics, attention methods, and also the transformer architectures in more depth as the risk of selection bias is by far the highest for this subgroup of methods. Among the attention methods, the relevance-filtered-based LA method scores primarily higher than non-filtered raw attention. In addition, LA allows to visualize input features that attribute to a specific outcome and are not only detected by the model in general, making it much more versatile.

**Compared to other method subgroups, SHAP methods differ strongly in performance.** Contrarily to other method subgroups such as linear surrogate methods (LIME, KS), CAM methods (GC, SC, C+), and attention methods (RA, RoA, LA), the Shapely value approximating SHAP methods (EG, KS, and DLS) differ extensively in their performance. This observation is consistent with the results of Molnar et al. [41], which are, however, not in the context of XAI evaluation. Therefore, it is advisable not to select a single SHAP method with the expectation of achieving similar results to others but rather to employ multiple such methods. For more results regarding behavioral similarities among XAI methods, we refer to Appendix M.

**For LRP we observe a trade-off between faithfulness and complexity.** In subsection 3.3 we already discussed our reservation against the complexity metrics and why especially CAM and attribution methods rank low. However, besides the attention methods, we also observe a strong trade-off between faithfulness and complexity for LRP, which we would also relate to the mathematical formulation of the complexity metrics. We can explain this observation by LRP's tendency to attribute to a very small set of input features: Faithfulness is low due to the absence of important input features in the attributed set, and robustness is low as the relative change in this set can occur fast, but complexity, as evaluated in our metrics, is also low due to the small set size. The attributed set size can be influenced by the model-layer assigned relevance propagation rule, e.g. switching the $\epsilon$-rule with the 0-rule, or hyperparameters, but this has to be fine-tuned per observation, making LRP only versatile when explaining single observations.

# 5 Comparison with related work

Our study addresses the limitations of previous research, which uses small and varying subsets of XAI methods and metrics, by providing a comprehensive and robust analysis that includes measurements of disagreement between metrics, thereby significantly enhancing validity and resolving gaps and inconsistencies between related studies. These gaps and inconsistencies become particularly evident from Table 1, which presents a summary of 18 related relevant studies (all on image data as there are none for volume or point cloud). While some "evergreen" XAI methods, i.e., VG, IxG, GB, and IG, stand out, the sparsity of Table 1 is very notable, especially for attention methods. In comparison, we present our results for the imaging modality indicated at the bottom, demonstrating the extensive difference in scale. Surprisingly, our consistently top-ranking method in terms of faithfulness and robustness, EG, is not evaluated in any of the related studies.

Back-referencing to Table 1, we observe in several cases similar results to other studies on image data: low faithfulness of VG, LIME, or LRP by Chefer et al. [15], and high faithfulness of IG (can depend highly on the selected baseline [6]) and LA (but only two studies including attention methods). Regarding the conflicting outcomes reported for GC, our results show average faithfulness but high robustness on image data (but can depend on the underlying model, as our work suggests). On the contrary, our results contradict the findings on high faithfulness and robustness of KS (Bhatt et al. [9] uses lower dimensional image data), high faithfulness of LRP, or low faithfulness of IG. However, these results can differ between modalities, as GC, for example, obtains very low scores in faithfulness and robustness on volume data. No related studies that examine both attention and attribution methods address the notably higher SD observed in attention methods compared to attribution methods when evaluating faithfulness.

Most evaluation of complexity is qualitative (e.g. Singh et al. [59]), with only those studies that introduce a metric themselves conducting also quantitative evaluations (i.e. Nguyen and Martínez [44], Kakogeorgiou and Karantzalos [31]). We consider the high fluctuation between quantitative and especially qualitative complexity evaluation outcomes as further support for our hypothesis that there is a gap between the aim of the metrics and the human conception of low complexity, strongly recommending the development of either new metrics or falling back to robust qualitative user studies.

# 6 Conclusion and discussion

Although our benchmark is one of the most comprehensive in the field, we restrict ourselves to the computer vision modalities with the, in our opinion, most unique and not overlapping characteristics, ignoring e.g. videos. Non-computer vision modalities, such as language, introduce new modality-specific XAI methods and metrics, rendering large-scale comparisons between these modalities infeasible. We also did not include more unconventional post-hoc XAI methods such as symbolic representations and meta-models or niche evaluation criteria like localization and axiomatic properties as they either require ground-truth bounding boxes or can not be applied to all XAI methods. Further, our benchmark focuses on the comparison between methods, not on the evaluation to what extent an individual method may or may not be faithful or robust in general, thus ignoring e.g. synthetic baselines.

Our results demonstrate vividly the need for rethinking the evaluation of XAI methods and the risks of inconsistent benchmarking for practitioners and researchers. As a solution, we offer practitioners profound benchmarking capabilities, practical takeaways for applying and selecting XAI methods, and adapted XAI methods and metrics for 3D modalities. This includes the most all-encompassing answer to *"What XAI method should I (not) use for my problem?"* to date, based on the extensive evidence in our provided result tables and the LATEC dataset. For researchers, we propose a new evaluation scheme, address the risk of conflicting metrics, and introduce LATEC as a platform for standardized benchmarking of methods and metrics in XAI. LATEC offers researchers the opportunity to explore and answer numerous critical questions in XAI, thereby playing a pivotal role in the advancement of the field.

**Acknowledgments**

This work was funded by Helmholtz Imaging (HI), a platform of the Helmholtz Incubator on Information and Data Science.

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

# Appendix

# A   Model performance and hyperparameter

## A.1   Test set performance

**a.**   Testset Performance on Image Modality

| Dataset: | Model Architecture: | Model Performance Metric | | | | |
|---|---|---|---|---|---|---|
| | | Accuracy | Precision | Recall | F1 | AUROC |
| OCT (MC: 4) | ResNet 50 | 0.999 | 0.999 | 0.999 | 0.999 | 1.0 |
| | EfficientNet b0 | 0.9969 | 0.9969 | 0.9969 | 0.9969 | 1.0 |
| | DeiT ViT | 0.999 | 0.999 | 0.999 | 0.999 | 1.0 |
| R45 (MC: 45) | ResNet 50 | 0.9535 | 0.9536 | 0.9538 | 0.9535 | 0.9995 |
| | EfficientNet b0 | 0.9554 | 0.9554 | 0.9549 | 0.9549 | 0.9995 |
| | DeiT ViT | 0.9568 | 0.957 | 0.9568 | 0.9567 | 0.9995 |

**b.**   Testset Performance on Volume Modality

| Dataset: | Model Architecture: | Model Performance Metric | | | | |
|---|---|---|---|---|---|---|
| | | Accuracy | Precision | Recall | F1 | AUROC |
| AMN (BC) | 3D ResNet 18 | 0.8003 | 0.8013 | 0.7987 | 0.8 | 0.8699 |
| | EfficientNet3D b0 | 0.8003 | 0.7954 | 0.8087 | 0.802 | 0.8647 |
| | Simple3DFormer | 0.7936 | 0.7907 | 0.7907 | 0.7907 | 0.8728 |
| OMN (MC: 11) | 3D ResNet 18 | 0.9115 | 0.9248 | 0.9248 | 0.9226 | 0.9953 |
| | EfficientNet3D b0 | 0.8754 | 0.8924 | 0.8936 | 0.8914 | 0.9893 |
| | Simple3DFormer | 0.8131 | 0.8463 | 0.8381 | 0.84 | 0.9815 |
| VMN (BC) | 3D ResNet 18 | 0.9359 | 0.937 | 0.9346 | 0.9358 | 0.98 |
| | EfficientNet3D b0 | 0.9162 | 0.9162 | 0.9162 | 0.9162 | 0.9229 |
| | Simple3DFormer | 0.8861 | 0.8871 | 0.8848 | 0.886 | 0.9394 |

**c.**   Testset Performance on Point Cloud Modality

| Dataset: | Model Architecture: | Model Performance Metric | | | | |
|---|---|---|---|---|---|---|
| | | Accuracy | Precision | Recall | F1 | AUROC |
| CMA (MC: 12) | PointNet | 0.9852 | 0.9743 | 0.9876 | 0.98 | 0.998 |
| | DGCNN | 0.9535 | 0.9373 | 0.9498 | 0.9423 | 0.9989 |
| | PC Transformer | 0.9751 | 0.9645 | 0.9688 | 0.9662 | 0.9996 |
| M40 (MC: 40) | PointNet | 0.8914 | 0.8374 | 0.8564 | 0.8438 | 0.9958 |
| | DGCNN | 0.9177 | 0.8844 | 0.891 | 0.8864 | 0.9973 |
| | PC Transformer | 0.9149 | 0.8779 | 0.8842 | 0.8796 | 0.9969 |
| SHN (MC: 16) | PointNet | 0.9878 | 0.9673 | 0.9689 | 0.9668 | 0.9991 |
| | DGCNN | 0.9903 | 0.966 | 0.9847 | 0.9745 | 0.9995 |
| | PC Transformer | 0.9896 | 0.9642 | 0.9819 | 0.9716 | 0.9997 |

**MC #**: Multi-Class (# Classes), **BC**: Binary-Class

Table 3: **a., b. & c.** Test set performance measured with the metrics: accuracy, precision, recall, F1, and area under the receiver operating characteristic (AUROC) curve, for each modality. In the case of IMN we use pretrained weights for the Transformer architecture from Huggingface[1] and the CNN architectures from TorchHub[2,3].

[1] `https://huggingface.co/facebook/deit-small-patch16-224`

[2] `https://pytorch.org/vision/stable/models/generated/torchvision.models.resnet50.html`

[3] `https://pytorch.org/vision/stable/models/generated/torchvision.models.efficientnet_b0.html`

Architectures were chosen based on their popularity and, to a limited extent, comparability between modalities, e.g. ResNet-50 and 3D ResNet-18 which both emerge from the same family of ResNet architectures. While 3D volume architectures could also be applied to point cloud data, we choose point cloud specific architectures for the modality. All models were trained on a NVIDIA GeForce RTX 3090.

## A.2 Hyperparameter

We tuned all hyperparameters on either the declared validation set or sampled a validation set based on 20% of the train set. The tuning was performed via grid search for each model. The primary metric for hyperparameter tuning was the F1 score.

**Utilized Image Datasets**

| Model Architecture | Hyperparameter | OCT | R45 |
|---|---|---|---|
| | Batch size | 128 | 128 |
| | Max Epochs | 8 | 60 |
| | Learning rate (LR) | 0.0001 | 0.0001 |
| | Optimizer | Madgrad | Madgrad |
| | LR Scheduler | Cosine Annealing | Cosine Annealing |
| | Weight Decay | 0 | 0 |
| | Momentum | 0.9 | 0.9 |
| ResNet 50 | Augmentations | Train:
Resize (256,256)
RandomCrop (224,224)
RandomAffine (shear=0.2, degrees=5)
RandomHorizontalFlip
Grayscale (channels=3)

Test:
Resize (256,256)
CenterCrop (224,224)
Grayscale (channels=3) | Train:
Resize (256,256)
RandomCrop (224,224)
RandomHorizontalFlip
RandAugment
Normalize (mean=(0.485, 0.456, 0. 406), std=(0.229, 0.224, 0.225))

Test:
Resize (256,256)
Normalize (mean=(0.485, 0.456, 0. 406), std=(0.229, 0.224, 0.225)) |
| | Sampling | Weighted Random Sampling | None |
| | Batch size | 128 | 128 |
| | Max Epochs | 5 | 15 |
| | Learning rate (LR) | 0.0001 | 0.001 |
| | Optimizer | Madgrad | Madgrad |
| | LR Scheduler | Cosine Annealing | Cosine Annealing |
| | Weight Decay | 0 | 0 |
| | Momentum | 0.9 | 0.9 |
| EfficientNet b0 | Augmentations | Train:
Resize (256,256)
RandomCrop (224,224)
RandomAffine (shear=0.2, degrees=5)
RandomHorizontalFlip
Grayscale (channels=3)

Test:
Resize (256,256)
CenterCrop (224,224)
Grayscale (channels=3) | Train:
Resize (256,256)
RandomCrop (224,224)
RandomHorizontalFlip
RandAugment
Normalize (mean=(0.485, 0.456, 0. 406), std=(0.229, 0.224, 0.225))

Test:
Resize (256,256)
Normalize (mean=(0.485, 0.456, 0. 406), std=(0.229, 0.224, 0.225)) |
| | Sampling | Weighted Random Sampling | None |
| | Batch size | 128 | 128 |
| | Max Epochs | 6 | 60 |
| | Learning rate (LR) | 0.0001 | 0.0001 |
| | Optimizer | Madgrad | Madgrad |
| | LR Scheduler | Cosine Annealing | Cosine Annealing |
| | Weight Decay | 0 | 0 |
| | Momentum | 0.9 | 0.9 |
| DeIT ViT | Augmentations | Train:
Resize (256,256)
RandomCrop (224,224)
RandomAffine (shear=0.2, degrees=5)
RandomHorizontalFlip
Grayscale (channels=3)

Test:
Resize (256,256)
CenterCrop (224,224)
Grayscale (channels=3) | Train:
Resize (256,256)
RandomCrop (224,224)
RandomHorizontalFlip
RandAugment
Normalize (mean=(0.485, 0.456, 0. 406), std=(0.229, 0.224, 0.225))

Test:
Resize (256,256)
Normalize (mean=(0.485, 0.456, 0. 406), std=(0.229, 0.224, 0.225)) |
| | Sampling | Weighted Random Sampling | None |

Table 4: Hyperparameter for all three architectures and CV datasets, excluding IMN as we load pretrained weights.

**Utilized Volume Datasets**

| Model Architecture | Hyperparameter | AMN | OMN | VMN |
|---|---|---|---|---|
| 3D ResNet18 | Batch size | 32 | 32 | 32 |
| | Max Epochs | 100 | 100 | 100 |
| | Learning rate (LR) | 0.001 | 0.001 | 0.001 |
| | Optimizer | SGD | Adam | Adam |
| | LR Scheduler | Cosine Annealing | Cosine Annealing | Cosine Annealing |
| | Weight Decay | 0 | 0 | 0 |
| | Momentum | 0.9 | 0 | 0 |
| | Augmentations | Train: RandomBrightness($U(0,1)$) Test: FixedBrightness(0.5) | None | Train: RandomBrightness($U(0,1)$) Test: FixedBrightness(0.5) |
| | Sampling | Weighted Random Sampling | None | Weighted Random Sampling |
| 3D EfficientNet b0 | Batch size | 32 | 32 | 64 |
| | Max Epochs | 100 | 100 | 100 |
| | Learning rate (LR) | 0.001 | 0.001 | 0.001 |
| | Optimizer | SGD | AdamW | Adam |
| | LR Scheduler | Cosine Annealing | Cosine Annealing | Cosine Annealing |
| | Weight Decay | 0.0005 | 0.0005 | 0 |
| | Momentum | 0.9 | 0 | 0 |
| | Augmentations | Train: RandomBrightness($U(0,1)$) Test: FixedBrightness(0.5) | None | Train: RandomBrightness($U(0,1)$) Test: FixedBrightness(0.5) |
| | Sampling | Weighted Random Sampling | None | Weighted Random Sampling |
| Simple3DFormer | Batch size | 32 | 32 | 64 |
| | Max Epochs | 150 | 100 | 100 |
| | Learning rate (LR) | 0.001 | 0.000001 | 0.001 |
| | Optimizer | SGD | Madgrad | Adam |
| | LR Scheduler | Cosine Annealing | Cosine Annealing | Cosine Annealing |
| | Weight Decay | 0.0005 | 0 | 0 |
| | Momentum | 0.9 | 0.9 | 0 |
| | Augmentations | Train: RandomBrightness($U(0,1)$) Test: FixedBrightness(0.5) | None | Train: RandomBrightness($U(0,1)$) Test: FixedBrightness(0.5) |
| | Sampling | Weighted Random Sampling | None | Weighted Random Sampling |

Table 5: Hyperparameters for all three architectures and CV datasets.

Utilized Point Cloud Datasets

| Model Architecture | Hyperparameter | CMA | M40 | SHN |
|---|---|---|---|---|
| PointNet | Batch size | 32 | 24 | 32 |
| | Max Epochs | 100 | 200 | 200 |
| | Learning rate (LR) | 0.001 | 0.001 | 0.001 |
| | Optimizer | AdamW | AdamW | AdamW |
| | LR Scheduler | Cosine Annealing | Cosine Annealing | Cosine Annealing |
| | Weight Decay | 0.0001 | 0.0001 | 0.0001 |
| | Momentum | 0 | 0 | 0 |
| | Augmentations | Pretransforms: NormalizeScale

Train: SamplePoints (1024) RandomScale (0.67,1.5) RandomRotate (degrees=15) RandomJitter (0.02)

Test: SamplePoints (1024) | Pretransforms: NormalizeScale

Train: SamplePoints (1024) RandomScale (0.67,1.5) RandomJitter (0.02)

Test: SamplePoints (1024) | Pretransforms: NormalizeScale

Train: RandomScale (0.67,1.5) RandomJitter (0.01) RandomRotate (degress=15, axis = (0,1,2))

Test: None |
| | Sampling | None | None | None |
| DGCNN | Batch size | 32 | 32 | 32 |
| | Max Epochs | 100 | 250 | 200 |
| | Learning rate (LR) | 0.001 | 0.001 | 0.001 |
| | Optimizer | AdamW | AdamW | AdamW |
| | LR Scheduler | Cosine Annealing | Cosine Annealing | Cosine Annealing |
| | Weight Decay | 0.0001 | 0.0001 | 0.0001 |
| | Momentum | 0 | 0 | 0 |
| | Augmentations | Pretransforms: NormalizeScale

Train: SamplePoints (1024) RandomScale (0.67,1.5) RandomRotate (degrees=15) RandomJitter (0.02)

Test: SamplePoints (1024) | Pretransforms: NormalizeScale

Train: SamplePoints (1024) RandomScale (0.67,1.5) RandomJitter (0.02)

Test: SamplePoints (1024) | Pretransforms: NormalizeScale

Train: RandomScale (0.67,1.5) RandomJitter (0.01) RandomRotate(degress=15, axis = (0,1,2))

Test: None |
| | Sampling | None | None | None |
| PC Transformer | Batch size | 32 | 32 | 32 |
| | Max Epochs | 150 | 250 | 200 |
| | Learning rate (LR) | 0.01 | 0.01 | 0.01 |
| | Optimizer | SGD | SGD | SGD |
| | LR Scheduler | Cosine Annealing | Cosine Annealing | Cosine Annealing |
| | Weight Decay | 0.0005 | 0.0005 | 0.0005 |
| | Momentum | 0.9 | 0.9 | 0.9 |
| | Augmentations | Pretransforms: NormalizeScale

Train: SamplePoints (1024) RandomScale (0.67,1.5) RandomRotate (degrees=15) RandomJitter (0.02)

Test: SamplePoints (1024) | Pretransforms: NormalizeScale

Train: SamplePoints (1024) RandomScale (0.67,1.5) RandomJitter (0.02)

Test: SamplePoints (1024) | Pretransforms: NormalizeScale

Train: RandomScale (0.67,1.5) RandomJitter (0.01) RandomRotate(degress=15, axis = (0,1,2))

Test: None |
| | Sampling | None | None | None |

Table 6: Hyperparameters for all three architectures and CV datasets.

## B   The LATEC dataset: Reference data for standardized evaluation

The resulting data of the three stages, which comprise the LATEC dataset, include pretrained model weights (excluding IMN), saliency maps, and evaluation scores. Thanks to the LATEC dataset, future experiments can start at a certain stage and use the results from the previous stage without recomputing everything again, e.g. when testing out a new evaluation metric on the existing saliency maps, preserving comparability. For the LATEC dataset, we compute per dataset saliency maps for the entire test set or 1000 observations depending on which size is smaller (on the validation set if the test set is unavailable), from which we sample 50 observations to compute evaluation scores for all 7,560 combinations. In total, the LATEC dataset consists of 326,790 saliency maps and 378,000 evaluation scores. As for such large datasets, the size can go into the hundreds of gigabytes. To save disk space, saliency maps could be cast from 64-bit precision to 32 or even 16-bit. We would, however, strongly advise against this, as even casting to 32-bit precision introduced numerical instability in our experiments due to the rounding of attribution and attention values, resulting in all-zero saliency maps and *nan* or *inf* evaluation scores. Further, as ranking lengths between CNN and Transformer architectures differ (attention methods only for Transformer architectures), we recompute rankings in the subsequent study, which aggregate over all three architectures by first combining the normalized evaluation scores per model architecture and then computing the ranking, preserving equal length between rankings (see Appendix F).

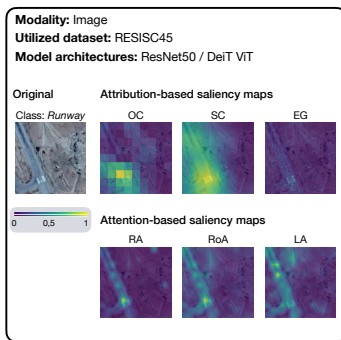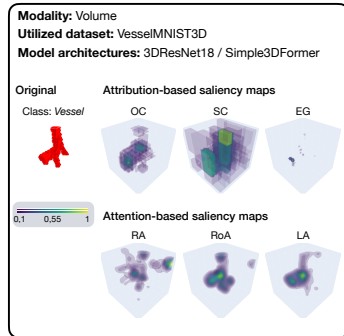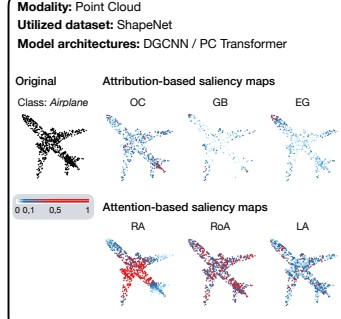

Figure 3: Illustrative saliency maps for all three modalities. The upper row shows three attributions, respectively, and the lower row, three attention-based methods. We observe how all XAI methods highlight the runway in the image and the vessel for the volume modality but with different granularity and focus. For the point cloud plane, explanations are less understandable, with attribution methods highlighting single points at the front tip, rudder, or wing tips.

To ensure a standardized setting with fair comparability between XAI methods over all possible experiment set-ups and aggregation levels, we take precautions regarding e.g. different types of feature attributions or the conversion of all metrics to single scores (see Appendix E for all detailed procedures). LRP requires non-negative activation outputs [43], leading us to a replacement of such activation functions (i.e. GeLU, leakyReLU) in CNN models, but we keep them for Transformer models, as they are central to the architecture and therefore also to our benchmark, and apply the $0^+$-rule instead.

## C  XAI methods overview and parameters

### C.1  Overview

#### C.1.1  Attribution Methods

**Occlusion [OC]** [72]    Systematically obscures different parts of the input data and observes the resulting impact on the output, to determine which parts of the data are most important for the model's predictions.

**LIME** [LIME] [50]    Creates an interpretable model around the prediction of a complex model to explain individual predictions locally (patch-based in our case), using perturbations of the input data and observing the corresponding changes in the output.

**Kernel SHAP [KS]** [39]    Using a weighted linear regression model as the local surrogate and selecting a suitable weighting kernel, the regression coefficients from the LIME surrogate can estimate the SHAP values.

**Vanilla Gradient [VG]** [58]    The raw input gradients of the model.

**Input x Gradient [IxG]** [57]    Multiples the input features by their corresponding gradients with respect to the model's output.

**Guided Backprob [GB]** [61]    Modifies the standard backpropagation process to only propagate positive gradients for positive inputs through the network, thereby creating visualizations that highlight the features that strongly activate certain neurons in relation to the target output.

**GradCAM [GC]** [56]    Uses the gradients of the target class flowing into the final convolutional layer to produce a coarse localization map by, highlighting the important regions in the image by up-scaling the map.

**ScoreCAM [SC]** [65]    Eliminates the need for gradient information by determining the importance of each activation map based on its forward pass score for the target class, producing the final output through a weighted sum of these activation maps.

**GradCAM++ [C+]** [14]    Generates a visual explanation for a given class label by employing a weighted sum of the positive partial derivatives from the final convolutional layer's feature maps, using them as weights with respect to the class score.

**Integrated Gradients [IG]** [62]    Explains model predictions by attributing the prediction to the input features, calculating the path integral of the gradients along the straight-line path from a baseline input to the actual input.

**Expected Gradients [EG]** [23]    Also called Gradient SHAP. Avoids the selection of a baseline value compared to IG, by leveraging a probabilistic baseline computed over a sample of observations.

**DeepLIFT [DL]** [57]    Assigns contribution scores to each input feature based on the difference between the feature's activation and a reference activation, effectively measuring the feature's impact on the output compared to a baseline.

**DeepLIFT SHAP [DLS]** [39]    Combines the DeepLIFT method with Shapley values to assign importance scores to input features by computing their contributions to the output relative to a reference input, while ensuring consistency with Shapley values.

**Layer-Wise Relevance Propagation [LRP]** [11]    Explains neural network decisions by backpropagating the output prediction through the layers, redistributing relevance scores to the input features to visualize their contribution to the final decision. We use the $\epsilon$-,$\gamma$- and $0^+$-rules depending on the model architecture for relevance backpropagation.

### C.1.2    Attention Methods

**Raw Attention [RA]** [22]    Rearranged and up-scaled attention values of the last attention head.

**Rollout Attention [RoA]** [1]    Averages attention weights of multiple heads to trace the contribution of each part of the input data through the network.

**LRP Attention [LA]** [15]    Assigns local relevance scores to attention weights based on the Deep Taylor Decomposition principle and propagates these relevancy scores through the model.

## C.2 Parameters

| Method | Hyperparameter | Image | Volume | Point Cloud |
|--------|----------------|-------|--------|-------------|
| OC | strides | 25 | 4 | 1 |
|  | sliding_window_shapes | (50,50) | (7,7,7) | (3,1) |
|  | baseline | 0 | 0 | 0 |
|  | perturbations_per_eval | 1 | 1 | 5 |
| LIME | alpha | 1 | 1 | 4 |
|  | n_samples | 10 | 10 | 10 |
|  | perturbations_per_eval | 5 | 5 | 5 |
| KS | baseline | 0 | 0 | 0 |
|  | n_samples | 10 | 10 | 10 |
|  | perturbations_per_eval | 5 | 5 | 5 |
| GC/SC/C+ | layer | ResNet50.layer4[-1] EfficientNetbo.features[-1] ViT.blocks[-1].norm1 | 3DEfficientNetbo.blocks[-13] 3DResNet18.layer3 S3DF.blocks[-1].norm1 | PointNet.transform.bn1 DGCNN.conv5 PCT.sa4.after_norm |
| SC | batch_size | 32 | 64 | 16 |
| IG | baseline | 0 | 0 | 0 |
|  | n_steps | 30 | 30 | 30 |
| EG | n_samples | 40 | 40 | 16 |
|  | std | 0.001 | 0.001 | 0.001 |
| DL | eps | 1E-09 | 1E-09 | 1E-09 |
|  | baseline | 0 | 0 | 0 |
| LRP | rule | $\varepsilon$ & $\gamma$-rule or $0^+$-rule | $\varepsilon$ & $\gamma$-rule or $0^+$-rule | $\varepsilon$ & $\gamma$-rule or $0^+$-rule |
|  | eps | 0,0001 | 0,0001 | 0,0001 |
|  | gamma | 0,25 | 0,25 | 0,25 |
| RA | layer | ViT.blocks[-1].attn | S3DF[-1].attn | PCT.sa4.attn |

Table 7: Parameters for each XAI method and modality.

The parameters for each XAI method are derived for each modality via qualitative evaluation which we deem the most realistic scenario. We tuned the XAI methods on five observations per dataset and modality, which we argue is a fair trade-off between fitting the methods to the dataset but not overfitting them to bias the evaluation. We did not tune the parameters per dataset, as the parameters transfer very well between datasets and only needed minimal adjustments. We computed all saliency maps on a compute cluster leveraging one NVIDIA A100 to compute saliency maps with a batch size of 10 for image and volume modality and 32 for point cloud modality.

# D   Evaluation metrics overview and parameters

## D.1   Overview

### D.1.1   Faithfulness

**FC** [9]    Gauges an explanation's fidelity to model behavior. It measures the linear correlation between predicted logits of modified test points and the average explanation for selected features, returning a score between -1 and 1. For each test, selected features are replaced with baseline values, and Pearson's correlation coefficient is determined, averaging results over multiple tests.

**FE** [5]     Evaluates the accuracy of estimated feature relevances by using a proxy for the "true" influence of features, as the actual influence is often unavailable. This is done by observing how the model's prediction changes when certain features are removed or obscured. Specifically, for probabilistic classification models, the metric looks at how the probability of the predicted class drops when features are removed. This drop is then compared to the interpreter's prediction of that feature's relevance. The metric also computes correlations between these probability drops and relevance scores across various data points.

**MC** [44]     Evaluates the correlation between the absolute values of attributions and the uncertainty in probability estimation using Spearman's coefficient. If attributions are not monotonic the authors argue that they are not providing the correct importance of the features.

**PF** [8]     The core concept involves flipping pixels with very high, very low, or near-zero attribution scores. The effect of these changes is then assessed on the prediction scores, with the average prediction being determined.

**RP** [55]     A step-by-step method where the class representation in the image, as determined by a function, diminishes as we gradually eliminate details from an image. This process, known as RP, occurs at designated locations. Finally, the effect on the average prediction is calculated.

**INS** [47]     Gradually inserts features into a baseline input, which is a strongly blurred version of the image, to not create OOD examples. During this process, the change in prediction is measured and the correlation with the respective attribution value is calculated.

**DEL** [47]     Deletes input features one at a time by replacing them with a baseline value based on their attribution score. During this process, the change in prediction is measured and the correlation with the respective attribution value is calculated.

**Iterative Removal of Features (IROF)** [51]     The metric calculates the area under the curve for each class based on the sorted average importance of feature segments (superpixels). As these segments are progressively removed and prediction scores gathered, the results are averaged across multiple samples.

**Remove and Debias (ROAD)** [52]     Evaluates the model's accuracy on a sample set during each phase of an iterative process where the k most attributed features are removed. To eliminate bias, in every step, the k most significant pixels, by the most relevant first order, are substituted with noise-infused linear imputations.

**Sufficiency** [18]     Assesses the likelihood that the prediction label for a specific observation matches the prediction labels of other observations which have similar saliency maps.

**Infidelity** [71]     Calculates the expected mean-squared error (MSE) between the saliency map multiplied by a random variable input perturbation and the differences between the model at its input and perturbed input.

### D.1.2    Robustness

**LLE** [5]     Lipschitz continuity in calculus is a concept that measures the relative changes in a function's output concerning its input. While the traditional definition of Lipschitz continuity is global, focusing on the largest relative deviations across the entire input space, this global perspective isn't always meaningful in XAI. This is because expecting consistent explanations for vastly different inputs isn't realistic. Instead, a more localized approach, focusing on stability for neighboring inputs, is preferred, resulting in a point-wise, neighborhood-based local Lipschitz continuity metric.

**MS** [71]     Measures the largest shift in the explanation when the input is slightly altered. It specifically evaluates the utmost sensitivity of a saliency map by taking multiple samples from a defined L-infinity ball subspace with a set input neighborhood radius, using Monte Carlo sampling for approximation.

**Continuity** [42]     Evaluates, that if two observations are nearly equivalent, then the explanations of their predictions should also be nearly equivalent. It then measures the strongest variation of the explanation in the input domain.

**Relative Input/Output/Representation Stability** [4]    All metrics leverage model information to evaluate the stability of a saliency map with respect to the change in the either, input data, intermediate representations, and output logits of the underlying prediction model.

### D.1.3 Complexity

**Sparseness** [12]    Measures the Gini Index on the vector of absolute saliency map values. The assessment ensures that features genuinely influencing the output have substantial contributions, while insignificant or only slightly relevant features should have minimal contributions.

**Complexity** [9]    Determines the entropy of the normalized saliency map.

**Effective Complexity** [44]    Evaluates the number of absolute saliency map values that surpass a threshold. Values above this threshold suggest the features are significant, while those below indicate they are not.

### D.2 Parameters

We tuned the parameters of the evaluation metrics per dataset based on the distribution of their scores. We applied the suggested parameters from [27] or the respective papers. If the resulting score distributions were collapsed, almost uniform, or too indistinguishable between the XAI methods, we tuned the respective parameters. This step was completed prior to the ranking analysis, and no adjustments were made to the metrics once the ranking phase commenced. We computed all evaluations on a compute cluster leveraging one NVIDIA A100 (40 GB VRAM) per dataset with a batch size of 2 (Batch size depends on the number of sampling steps of some metrics. See *nr_samples* in Table 8 for our number of samples per metric.)

| Evaluation Metric: | Hyperparameter: | Image | | | Volume | | | Point Cloud | | |
|---|---|---|---|---|---|---|---|---|---|---|
| | | IMN | OCT | R45 | AMN | OMN | VMN | CMA | M40 | SHN |
| FC | nr_runs | 100 | 100 | 100 | 100 | 100 | 100 | 100 | 100 | 100 |
| | subset_size | 224 | 224 | 224 | 56 | 56 | 56 | 32 | 32 | 32 |
| | perturb_baseline | black | black | black | black | black | black | center | center | center |
| FE | features_in_step | 224 | 224 | 224 | 56 | 56 | 56 | 32 | 32 | 32 |
| | perturb_baseline | black | black | black | black | black | black | center | center | center |
| MC | nr_samples | 10 | 10 | 10 | 10 | 10 | 10 | 10 | 10 | 10 |
| | features_in_step | 3136 | 3136 | 3136 | 392 | 392 | 392 | 256 | 256 | 256 |
| | perturb_baseline | uniform | uniform | uniform | uniform | uniform | uniform | uniform | uniform | uniform |
| PF | features_in_step | 224 | 224 | 224 | 56 | 56 | 56 | 32 | 32 | 32 |
| | perturb_baseline | black | black | black | black | black | black | center | center | center |
| RP | patch_size | 14 | 14 | 18 | 4 | 4 | 4 | 3 | 3 | 3 |
| | regions_evaluation | 10 | 10 | 20 | 20 | 20 | 20 | 32 | 32 | 32 |
| | perturb_baseline | uniform | uniform | uniform | uniform | uniform | uniform | uniform | uniform | uniform |
| INS | pixel_batch_size | 50 | 50 | 50 | 50 | 50 | 50 | 50 | 50 | 50 |
| | sigma | 5.0 | 120.0 | 40.0 | 2.5 | 2.5 | 2.5 | 0.05 | 0.1 | 0.05 |
| | kernel_size | 15 | 39 | 19 | 1 | 1 | 1 | 1 | 1 | 1 |
| DEL | pixel_batch_size | 50 | 50 | 50 | 50 | 50 | 50 | 50 | 50 | 50 |
| IROF | segmentation | Slic | Slic | Slic | 3D Slic | 3D Slic | 3D Slic | KMeans | KMeans | KMeans |
| | perturb_baseline | mean | mean | mean | black | black | black | center | center | center |
| ROAD | noise | 0.1 | 0.1 | 0.1 | 4.0 | 2.5 | 50.0 | 0.02 | 0.15 | 0.3 |
| | percentages_max | 100 | 100 | 100 | 100 | 100 | 100 | 100 | 100 | 100 |
| SUF | threshold | 0.9 | 0.6 | 0.6 | 0.02 | 0.75 | 0.0002 | 0.75 | 0.75 | 0.6 |
| LLE | nr_samples | 5 | 5 | 5 | 10 | 10 | 10 | 5 | 5 | 5 |
| | perturb_std | 0.1 | 0.0002 | 0.1 | 0.2 | 0.2 | 0.2 | 0.1 | 0.1 | 0.1 |
| | perturb_mean | 0.0 | 0.0 | 0.0 | 0.0 | 0.0 | 0.0 | 0.0 | 0.0 | 0.0 |
| MS | nr_samples | 10 | 10 | 10 | 10 | 10 | 10 | 10 | 10 | 10 |
| | lower_bound | 0.2 | 0.2 | 0.2 | 0.2 | 0.2 | 0.2 | 0.2 | 0.2 | 0.2 |
| CON | patch_size | 56 | 56 | 56 | 7 | 7 | 7 | 3 | 3 | 3 |
| | nr_steps | 20 | 20 | 20 | 20 | 20 | 20 | 20 | 20 | 20 |
| | perturb_baseline | uniform | uniform | uniform | uniform | uniform | uniform | uniform | uniform | uniform |
| RIS | nr_samples | 10 | 10 | 10 | 10 | 10 | 10 | 10 | 10 | 10 |
| ROS | nr_samples | 10 | 10 | 10 | 10 | 10 | 10 | 10 | 10 | 10 |
| RRS | nr_samples | 10 | 10 | 10 | 10 | 10 | 10 | 10 | 10 | 10 |
| INF | n_perturb_samples | 50 | 50 | 50 | 50 | 50 | 50 | 50 | 50 | 50 |
| ECP | eps | 0.01 | 0.01 | 0.01 | 0.001 | 0.001 | 0.001 | 0.001 | 0.001 | 0.001 |

Table 8: Parameters for all evaluation metrics on each CV dataset.

## D.3 Adaption of XAI methods

In this section, we explain how we adapted XAI methods in our framework to seamlessly work with 3D modalities. We neglect the methods that did not need any adaption (besides e.g. unit tests etc.) as they work independently of the input dimensions. All XAI methods are adapted, such that they only return positive attribution.

**Occlusion** For the 3D modalities we implemented a 3D kernel as the perturbation baseline for volumes and a 1x3 mask (one point) for the point clouds. The image and volume mask transverse with overlap and the point cloud mask without overlap over all dimensions of the input object.

**LIME & Kernel SHAP** For both methods, we implemented feature masks for each modality, as training the linear surrogate models on the original input features is not informative and computa-

tionally very expensive. Each mask groups the input features to the same interpretable feature. We use predefined grids as feature masks, as superpixel computing algorithms are too computational and time-expensive, especially for 3D modalities and evaluation metrics that perturb the input space or refit the XAI method multiple times. For the image modality, we use a 16x16x3, for volume 7x7x7, and for point cloud 1x3 (one point) mask, which is distributed as a non-overlapping grid in all dimensions over the whole object. For point clouds we use ridge regression and for the other modalities lasso regression.

**GradCAM, ScoreCAM & GradCAM++**    For all CAM methods on volume data we adapted the gradient averaging and the subsequent weighting of the activations and used nearest-neighbor interpolation to upscale the weighted activations to 3D volumes. In the case of ScoreCAM we also use nearest neighbor up-sampling instead of bilinear up-sampling, to upscale the activations for weighting the output of the previous layer. To correctly reshape the upscaled images and volumes in the case of the Transformer architectures (taking the channels to the first dimension as for CNNs), we use two different reshape functions for images and volumes when the CAM methods are applied to Transformer architectures. Further, we use the absolute activation output, not the non-negative for Transformer architectures, as the leaky-ReLU/GeLU function output otherwise would sometimes be zero.

**LRP**    For CNNs, we assigned the $\epsilon$-rule to the linear or identity layers, the identity rule to all non-linear layers, and to all other layers (convolutions, pooling, batch normalization, etc.) the $\gamma$-rule. For Transformer architectures we implemented the $0^+$-rule for all layers. However, for the Simple3DFormer and the PC Transformer, we had to add custom relevance propagation through the whole model, as the architectures come with several sub-modules such as "local gathering" for the PC Transformer, which are non-trivial to backpropagate through.

**Raw Attention**    We always use the raw attention of the last Transformer block and use bilinear or trilinear interpolation to rescale the attention for image and volume data. For point cloud data, this procedure is more complicated as the PC Transformer projects the embeddings on which the Transformer acts via farthest point sampling and k-nearest neighbor grouping. Thus in each downsampling step, we save which k points are sampled to then use k-nearest neighbor interpolation to cast the attention values for these remaining points back into the input space onto all 1024 original points.

**Rollout Attention**    Same procedure as for Raw Attention but before we interpolate back into the original input space, we use the rollout attention aggregation algorithm over all Transformer modules in the architecture.

**LRP Attention**    As for LRP we use custom relevance backpropagation for the Simple3DFormer and PC Transformer architectures. Based on the relevance scores, we filter the attention of each Transformer module, aggregate the filtered attention with the rollout algorithm, and interpolate the resulting attention back into the input as described for Raw Attention.

### D.4    Adaption of evaluation metrics

In this section, we explain how we adapted the evaluation metrics in our framework to seamlessly work with 3D modalities. All metrics were adapted for point cloud (n,d) and volume (x,y,z) dimensions besides classical image dimensions (w,h,c). We neglected the metrics which did not need any further adaption. All metrics leveraging threshold values expect normalized saliency maps on the observation level. Otherwise, thresholds have to be selected per observation.

**PF**    We compute the Area Under the Curve (AUC) to receive a single score. For point cloud data acts on the single coordinates.

**RP**    We compute the AUC to receive a single score. Acts on a 3D kernel for volume data and single points for point cloud data. Compute the AUC to receive a single score.

**INS**    Use Gaussian noise for 3D data instead of Gaussian blur for images. Inserting single points for point cloud data and voxels for volume data.

**DEL**    Deletes single points for point cloud data and voxels for volume data. Compute the AUC to receive a single score.

**Iterative Removal of Features (IROF)**    Compute the Area Over the Curve (AOC) to receive a single score. We use 3D Slic for volume segmentation and KMeans clustering with fixed $k = 16$ clusters for point cloud segmentation. $k = 16$ was determined by visual inspection. See exemplary visualization in Figure 4.

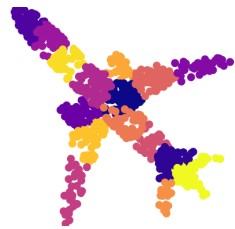

Figure 4: Example of KMeans clustering for point cloud data with k=16.

**Remove and Debias (ROAD)**    We use Gaussian noise for 3D modalities. Compute the AUC to receive a single score.

**Sufficiency**    Use the whole set of saliency maps for similarity comparison and not only the batch the metric is applied to (see Appendix L). For distance calculation between saliency maps, we use squared Euclidean distance for volume data and standardized Euclidean distance for image and point cloud data due to numerical instability.

**Continuity**    We implemented x-axis traversal for volume data along the x-axis with black padding in all dimensions and for the point cloud data by traversing all points along the x-axis position at $(n, d = 0)$ (see Figure 5). As removing points for point cloud data would change the input dimension of the object, we instead map them to the center (0,0,0). We did not observe any OOD behavior by implementing this solution. We use the Pearson Correlation Coefficient (PCC) between traversals to compute a single score.

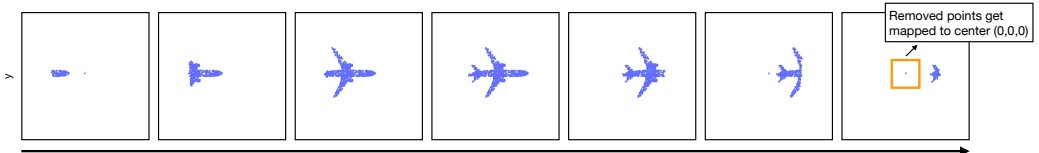

Figure 5: X-axis traversal of point clouds for continuity metric. We can not remove points as this would change the input dimensionality, thus we map them to the center (0,0,0), which is similar to black padding for image and volume data.

**Relative Representation Stability**    We use uniform noise ($U(0, 0.05)$) due to numerical stability as Gaussian noise could generate infinity values.

## E    Ensuring comparability of results

To ensure fair comparability between XAI methods over all possible experiment set-ups and aggregation levels, we take precautions about the XAI methods, evaluation metrics, and model architectures. Attribution measures the positive or negative contribution of an input feature (e.g. pixel) into the predicted output class of the model. On the contrary, CAM methods only compute positive attribution, and attention highlights all general (or absolute) important input features independent of the output class. However, in practice, attention is only valuable in interpretation if it also highlights features that are used for prediction. New methods such as LA filter the attention to only show such class-relevant attention, and their possible better performance to unfiltered attention can only be shown by evaluating it as positive attribution. Thus we consider only positive attribution for saliency map comparison (also suggested by Zhang et al. [73]).

Further, we normalize the saliency maps on the observation level as some metrics have nominal thresholds or noise intensities which depend on the scale of saliency maps. As not all metrics compute single scores we have to convert all metrics computing sequences or array of sequences into single scores either via the AUC for PF, RP, Selectivity and ROAD, AOC for IROF, or the PCC for SensitivityN and Continuity. All scores are normalized on the metric and dataset level. Score backpropagation-based metrics such as LRP (excluding the $0^+$-rule), DS or DLS, and the CAM methods expect non-negative activation outputs. Thus, we exchanged before the CNN model training all GeLU or leakyReLU activation functions with standard ReLU functions as they output negative values, biasing the XAI method. For the Transformer architectures, however, we keep all activation functions, as well as the skip connections and patchification, as they are central to the architecture. Their potential effect on different attribution methods is part of the benchmark. For CAM methods on the Transformer architectures, we interpolate the reshaped *absolute* cls token, as saliency maps would otherwise often be empty (also recommended by Chefer et al. [15]).

## F    Ranking computation flow chart

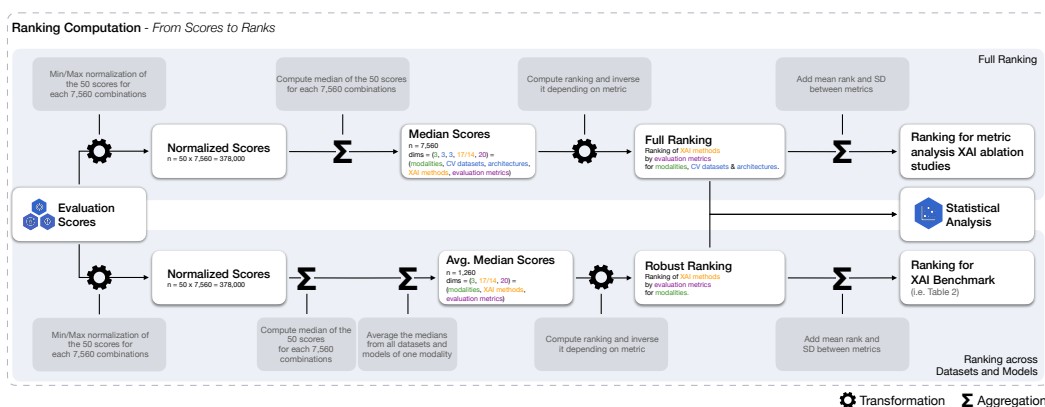

Figure 6: Transformation and aggregation steps from raw evaluation scores to final rankings.

Figure 6 shows the transformation and aggregation steps from raw scores to final rankings depending if we want to analyze the full ranking or achive a more robust ranking by averaging first across the median scores of dataset and model architecture combinations of each modality. In the calculation of the combinations, it must be taken into account that in the case of the Transformer architectures we have three more XAI methods (attention methods), and in the case of the point cloud modality, we have three fewer XAI methods (excluding CAM methods). In the case of the full ranking, we then have 7,560 combinations of input datasets, architectures, XAI models, and evaluation metrics based on which we compute 50 scores for each combination, but always the same observations per dataset. If we average medians of the score distributions of each architectures and datasets, we end up with 1,260 combinations but $6 * 50$ scores per combination, which are in total again 378,000 scores.

## G    Mathematical notation and formulas

We introduce a mathematical notation for subsequent calculations and formulas. Here, the parameters are represented as : modality $M$, utilized dataset $D$, model architectures $A$ (CNNs as $A_{C=\{c_1,c_2\}}$ and Transformer as $A_T$), XAI method $D$ (attribution methods as $F_A$ and attention methods as $F_T$) and evaluation metrics and criteria as $E$ ($E_F$ for faithfulness, $E_R$ for robustness and $E_C$ for complexity sets of metrics). The Rankings are denoted by $R$.

### G.1    Average SD between metrics (Figure 2 (b.))

We computed the standard deviation (SD) between metric rankings for each modality, criteria and model architecture ($\bar{\sigma}_{m,C_i,a}$), mean aggregated over XAI methods and datasets, as well as for each modality, criteria and dataset ($\bar{\sigma}_{m,C_i,d}$), mean aggregated over XAI methods and model architecture:

Using the notation introduced in Appendix G and SD as $\sigma(\cdot)$. Per model architecture $(a)$:

$$\bar{\sigma}_{m,C_i,a} = \frac{1}{|D||F|} \sum_{d \in D, f \in F} \sigma(R_{C_i,a,f,m,d}) \quad \forall m \in M, \ \forall a \in A, \ \forall C_i \in \{C_F, C_R, C_C\},$$

(1)

and dataset $(d)$:

$$\bar{\sigma}_{m,C_i,d} = \frac{1}{|A||F|} \sum_{a \in A, f \in F} \sigma(R_{C_i,a,f,m,d}) \quad \forall m \in M, \ \forall d \in D, \ \forall C_i \in \{C_F, C_R, C_C\}$$

(2)

## G.2 Proportion of accepted Levene tests (Figure 2 (c.))

Proportion of accepted tests $(\bar{\rho}_{m,C_i,f})$ of each criteria and XAI method, i.e. tests where the computed p-value — the probability that an observed effect occurs by chance — is below the significance level $\alpha$:

With significance level $\alpha = 0.1$, p-value $PV_{\text{Le}}(\cdot)$, indicator function $\mathbb{1}[\cdot]$ and variance $\sigma^2(\cdot)$.

$$\bar{\rho}_{m,C_i,f} = \frac{1}{|D||A|} \sum_{d \in D, a \in A} \mathbb{1}[PV_{\text{Le}}(\sigma^2(R_{C_i,f,m,a,d})) < \alpha]$$

(3)

$$\forall m \in M, \ \forall f \in F, \ \forall C_i \in \{C_F, C_R, C_C\}$$

(4)

## G.3 Average absolute rank distance between model architectures (Figure 10)

Distance of ranks between each model architecture for each attribution method and modality:

$$\bar{\delta}^1_{m,f,A_{ij}} = \frac{1}{|D||C|} \sum_{d \in D, c \in C} |R_{m,c,f,a_i} - R_{m,c,f,a_j}|$$

(5)

$$\forall m \in M, \ \forall c \in C, \ \forall f \in F, \ \forall \{a_i, a_j \neq a_i\} \in A$$

(6)

## G.4 Average Euclidean distance between metrics (Figure 8)

Average Euclidean ranking distance $(\bar{\bar{\delta}}_{m,c})$ between all metrics across image datasets and model architectures:

$$\bar{\bar{\delta}}_{m,c} = \frac{1}{|D||A||F|} \sum_{d \in D, a \in A, f \in F} \sqrt{(R_{m,c,d,a,f} - R_{m,c,d,a,f})^2} \quad \forall m \in M, \ \forall c \in C$$

(7)

## G.5 Ranking correlation between XAI methods (Figure 12 (a.)

Correlation in ranking between XAI methods, indicating their relative similarity.

With Pearson correlation coefficient $PCC$.

$$PCC_{F_{ij}} = PCC(R_{\bar{C},M,D,A,f_i}, R_{\bar{C},M,D,A,f_j}) \quad \forall f_i, f_j \in F$$

(8)

## H  Adapting current XAI methods and evaluation metrics for 3D data

While many XAI methods and evaluation metrics are independent of the input space dimensions, especially methods leveraging perturbations, interpolations for up- and down-scaling or segmentation are not. Our implementation builds upon the work from Kokhlikyan et al. [34] and Hedström et al. [28] for XAI methods and evaluation metrics for 1D and 2D images, and we extended it to 3D volume and point cloud data. Both modalities come with their own specifies, e.g. that local neighborhoods have to be defined via k-nearest neighbors (KNN) in point cloud data and not 2D or 3D patches as in image or volume data. For the XAI methods, we advanced e.g. OC, LI, and KS by the adoption of 3D patches, all three CAM methods with 3D interpolation, all attention-based methods with 3D and KNN-based interpolations, and LA with relevance backpropagation for the Simple3DFormer and PC Transformer architectures. As the adoption of the CAM methods for point cloud data and more complex architectures than PointNet is not trivial, we deem it out of scope for this paper and do not include them in our point cloud experiments. In the case of evaluation metrics, we adapted e.g. perturbation applying metrics to 3D patches or point-based perturbations, the superpixel segmentation in IROF by 3D Slic and KMeans clustering and padded x-axis transversal for the volume and point cloud data in Continuity. Additionally, we modified all methods and metrics to function with $(x, y, z)$ volume and $(n, 3)$ point cloud dimensions. All adaptations were tested for their coherency, and illustrative saliency maps can be observed in Figure 3. We refer to Appendix C and Appendix subsection D.2 for all implementation details.

## I  Metric standard deviation for volume and point cloud data

**a.**  Avg. Standard Deviation for Volume Model Architectures and Datasets

| Evaluation Criteria: | Model Architectures | | | Utilized CV Datasets | | |
|---|---|---|---|---|---|---|
| | 3DResNet18 | 3DEffNetb0 | S3DF | AMN | OMN | VMN |
| **Faithfulness** | 3.07 | 3.41 | 3.61 | 3.34 | 3.19 | 3.55 |
| **Robustness** | 3.47 | 3.41 | 3.42 | 3.54 | 3.25 | 3.51 |
| **Complexity** | 0.45 | 0.48 | 0.64 | 0.51 | 0.63 | 0.43 |
| *Weighted Average* | 2.82 | 2.99 | 3.1 | 2.99 | 2.83 | 3.07 |

**b.**  Avg. Standard Deviation for PC Model Architectures and Datasets

| Evaluation Criteria: | Model Architectures | | | Utilized CV Datasets | | |
|---|---|---|---|---|---|---|
| | PointNet | DGCNN | PCT | CMA | M40 | SHN |
| **Faithfulness** | 2.9 | 2.97 | 3.55 | 3.23 | 3.07 | 3.1 |
| **Robustness** | 2.52 | 2.74 | 2.91 | 2.9 | 2.8 | 2.48 |
| **Complexity** | 0.72 | 0.51 | 0.29 | 0.42 | 0.59 | 0.51 |
| *Weighted Average* | 2.44 | 2.52 | 2.84 | 2.69 | 2.6 | 2.49 |

Table 9: Average metric standard deviation per model architectures and utilized datasets for **a.** volume and **b.** point cloud modalities.

## J  Why do metrics disagree?

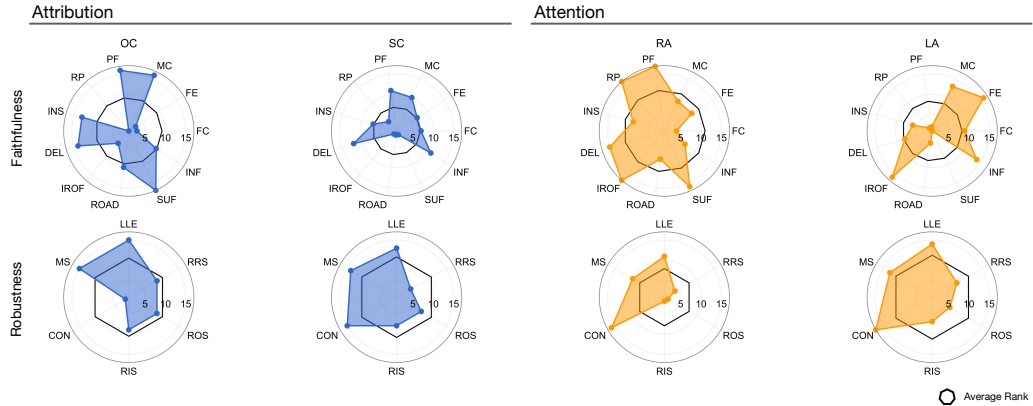

Figure 7: Differences between faithfulness and robustness metrics in ranking OC, SC, RA and LA. The black circle indicates the overall average rank.

We have established that all metrics approximate similar criteria, differing primarily in their interpretation and mathematical formulation. Although these differing perspectives mainly agree with their

rankings, our study reveals that variations in mathematical formulation can significantly contribute to metric disagreement. Our prior research indicates that the extent of disagreement between metrics is significantly influenced by the chosen XAI method. This dependency emerges as a critical factor in why certain metrics may favor or disadvantage specific XAI methods due to their mathematical structures. Moreover, our further experiments demonstrate that metrics particularly diverge in their rankings of XAI methods in terms of faithfulness, especially when the mechanism used for evaluation and the mechanism for computing the explanation (i.e. saliency map) are closely related.

In Figure 2 (c.) we can observe for OC that the SD between faithfulness ranks is never significantly smaller than the deviation of a random ranking (see exemplary Figure 7 for all individual faithfulness metric ranks of OC for one set of design parameters). A primary cause of this notable metric disagreement is OC's alignment with the RP metric. Both OC and RP involve perturbing a larger region of the input image with a baseline value, utilizing either a fixed kernel in OC or a set of pixels determined by ordered attribution values in RP. When the set size for RP matches the OC kernel size, and there is no overlap of the sliding kernels, the metric operationally mirrors the XAI method. However, our study indicates that even when the set size does not align, the metric still tends to favor the method. Figure 7 highlights OC's high rankings when evaluated using RP and lower rankings with metrics that employ finer, incremental pixel-level perturbations, such as PF, INS, or DEL. This evaluation bias stems from OC's inherent limitation of attributing to entire regions rather than individual pixels.

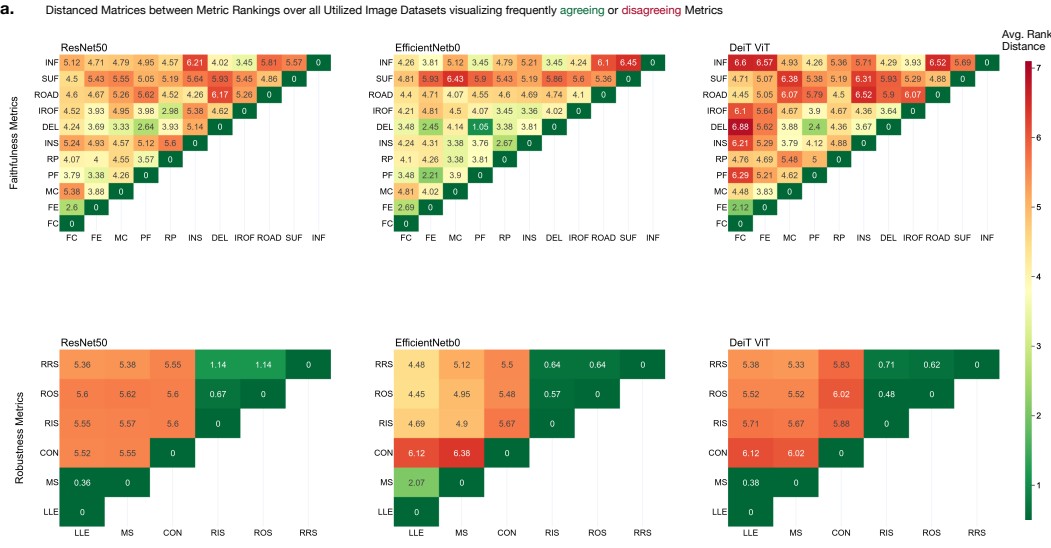

Figure 8: Average Euclidean ranking distance between metric pairs for model architectures and the faithfulness and robustness criteria. More often agreeing metric pairs in their rankings appear more green, and disagreeing pairs more red (see Equation 7).

We observe a dependency not only between specific XAI methods and metrics but also among metrics that utilize similar evaluation mechanisms or those designed to specifically address the shortcomings of other metrics. This scenario poses a risk for selection bias, as we observe that several related studies, such as Li et al. [38], predominantly select such metrics with similar methodologies and consequently similar ranking behavior.

Figure 8 illustrates the average Euclidean ranking distance ($\bar{\delta}_{m,c}$) between all metrics across image datasets and model architectures based on Equation 7. In assessing faithfulness, metrics that involve incremental pixel perturbation (PF, INS, DEL) and those that correlate attribution values with predicted logits (FC, FE) tend to rank more similarly. Conversely, metrics designed to mitigate specific limitations of these methods display notably different rankings. For instance, incrementally inserting or deleting pixel-based methods may generate out-of-distribution examples for the model, leading to highly uncertain or even random predictions. This issue is addressed by the ROAD metric,

which consequently ranks distinctly from DEL and PF. Similarly, the MC metric, which evaluates the correlation between the absolute attribution values and the uncertainty in probability estimates, addresses shortcomings in FC and FE, leading to divergent rankings. Regarding robustness metrics, variations are more consistent. There is a notable similarity in rankings between the LLE and MS, both of which measure relative changes when inputs are slightly altered, as well as between the Relative Stability metrics. Disparities in rankings among metrics may also arise from variations in the tuning of hyperparameters.

We additionally perform the same analysis through the correlation between metrics. While Euclidean distance and Pearson correlation both measure similarity, correlation focuses on trends, whereas distance measures actual differences. When assessing faithfulness in Figure 9, we find that metrics involving incremental pixel perturbation (PF, INS, DEL) and those correlating attribution values with predicted logits (FC, FE) are positively correlated. However, metrics specifically designed to address limitations in these methods, such as ROAD (which addresses the out-of-distribution issue in pixel perturbation) and MC (which incorporates uncertainty rather than logits), are interestingly even negatively correlated with them. Regarding robustness, the relative stability metrics show a positive correlation. In general the results are similar to the findings based on the distance matrices.

In summary, our study demonstrates that the variation in metric rankings for a given XAI method can be attributed to the similarity or dissimilarity in the mathematical mechanisms employed by the metrics themselves, as well as between the metrics and the underlying XAI method. However, in scenarios where there is substantial disagreement among metrics, selection biases may emerge if only a limited subset of metrics—those that potentially employ similar mechanisms—is considered. This underscores the importance of incorporating a diverse array of metrics to ensure an accurate approximation of a criterion, independent of the mathematical mechanisms involved.

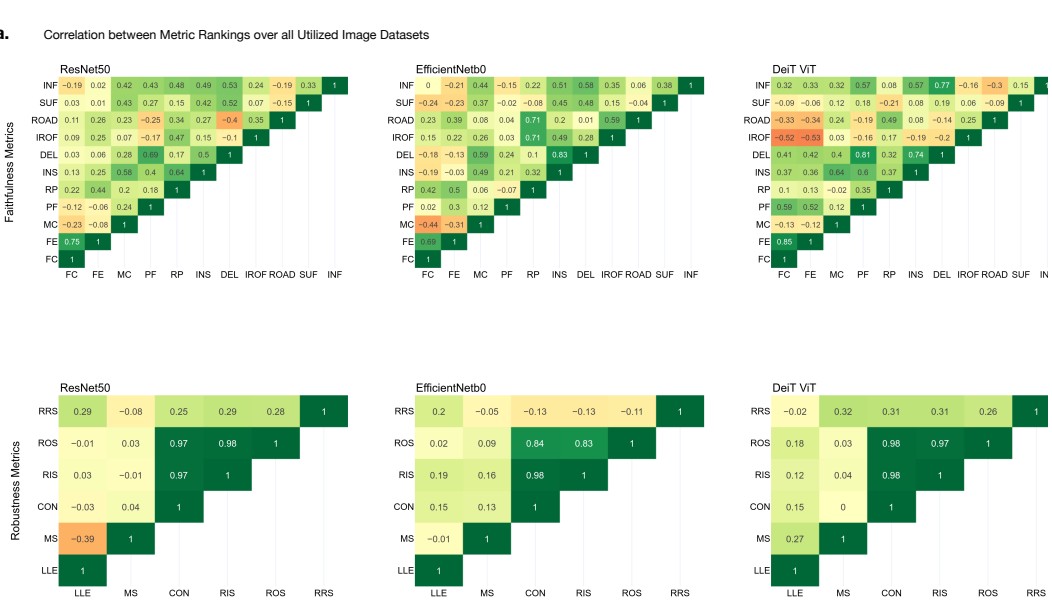

Figure 9: Correlation between metrics for model architectures and the faithfulness and robustness criteria. Positive correlation in green, negative in red.

## K Differences in ranking between datasets and model architectures.

### K.1 Ranking table across datasets

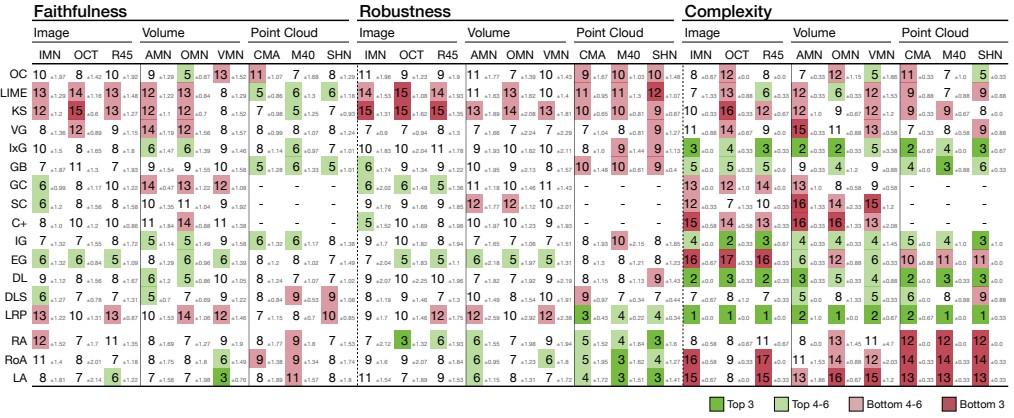

Table 10: Full ranking table for all XAI methods and CV datasets with standard error (SE).

When comparing the average metric rank between datasets belonging to one modality we observe only minor differences.

### K.2 Ranking distance between model architectures

Recent research around Vision-Transformers has shown distinct differences in their learning dynamics [48, 46], robustness [74] or latent representations [67] compared to classical CNNs. As many of their mechanisms (e.g., global processing, lack of inductive biases, (self-)attention, negative activations) can theoretically also affect attribution methods, we want to test if similar distinctions between Transformer and CNN architectures can also be detected for the performance of attribution methods. To this end, we analyze the distance of ranks between each model architecture for each attribution method in Figure 10 based on Equation 5.

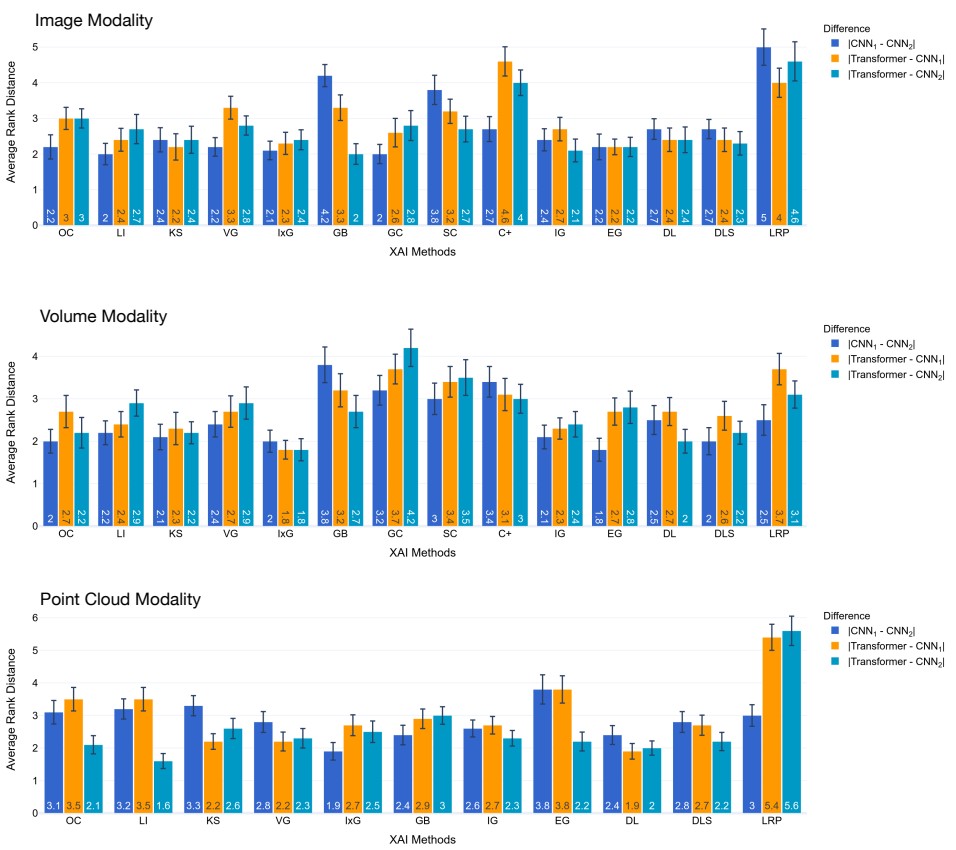

Figure 10: Average distance between ranks of XAI methods on different model architectures for all modalities.

The average rank distance $(\bar{\delta}^1_{m,f,A_{ij}})$ does not change substantially between CNN and Transformer architectures for most attribution methods (except for CAM methods and LRP), which is mainly centered around the mean value of the respective modality. This indicates that almost all attribution methods do not receive significantly different evaluation scores depending on the underlying architecture and we can not support the hypothesis that attribution methods behave fundamentally differently on Transformer architectures compared to CNNs. Two notable outliers are the CAM methods and LRP which show genrally higher inter rank distance. Also GB has higher rank distance between CNNs on the image and volume modality and EG on the point cloud modality.

## K.3 Ranking tables for CNN and Tranformer only

| Evaluation Criteria: | Faithfullness | | | Robustness | | | Complexity | | |
|---|---|---|---|---|---|---|---|---|---|
| Modality: | Image | Volume | Point Cloud | Image | Volume | Point Cloud | Image | Volume | Point Cloud |
| OC | 8.5 | 5.5 | 9 | 11.5 | 5.5 | 4.5 | 9 | 10.5 | 7.5 |
| LIME | 14 | 9.5 | 2.5 | 14 | 12.5 | 11 | 6 | 8.5 | 7.5 |
| KS | 13 | 9.5 | 7 | 13 | 14 | 9 | 13.5 | 8.5 | 10 |
| VG | 11.5 | 11 | 9 | 6 | 2 | 2 | 10 | 13 | 9 |
| IxG | 10 | 2.5 | 5 | 11.5 | 9.5 | 7 | 3.5 | 1.5 | 3 |
| GB | 8.5 | 7 | 1 | 10 | 8 | 8 | 2 | 5 | 5.5 |
| GC | 3 | 14 | - | 2 | 11 | - | 12 | 7 | - |
| SC | 5 | 8 | - | 8 | 9.5 | - | 7.5 | 14 | - |
| C+ | 4 | 13 | - | 4.5 | 7 | - | 11 | 12 | - |
| IG | 6 | 1 | 5 | 8 | 3 | 4.5 | 5 | 4 | 4 |
| EG | 1 | 5.5 | 11 | 2 | 1 | 10 | 13.5 | 10.5 | 11 |
| DL | 7 | 2.5 | 2.5 | 8 | 4 | 4.5 | 1 | 3 | 2 |
| DLS | 2 | 4 | 9 | 4.5 | 5.5 | 1 | 7.5 | 6 | 5.5 |
| LRP | 11.5 | 12 | 5 | 2 | 12.5 | 4.5 | 3.5 | 1.5 | 1 |

Per modality: ■ Top 1 □ Top 2-4 ▨ Bottom 2-4 ■ Bottom 1

Table 11: Average ranking of the CNN architectures. Coloring coincides with top and bottom positions as no attention methods can be applied to CNN architectures.

| Evaluation Criteria: | Faithfullness | | | Robustness | | | Complexity | | |
|---|---|---|---|---|---|---|---|---|---|
| Modality: | Image | Volume | Point Cloud | Image | Volume | Point Cloud | Image | Volume | Point Cloud |
| OC | 13 | 9 | 11.5 | 12.5 | 12 | 13 | 11.5 | 8.5 | 6.5 |
| LIME | 16.5 | 15 | 2 | 15.5 | 16 | 14 | 9 | 8.5 | 5 |
| KS | 15 | 14 | 4.5 | 17 | 17 | 10 | 17 | 12.5 | 9 |
| VG | 11.5 | 11 | 8.5 | 1.5 | 2 | 7 | 13.5 | 10 | 6.5 |
| IxG | 9.5 | 3 | 1 | 7.5 | 10 | 10 | 1 | 1 | 3 |
| GB | 5 | 5 | 3 | 10 | 13 | 12 | 4.5 | 5 | 4 |
| GC | 9.5 | 16.5 | - | 12.5 | 10 | - | 10 | 14 | - |
| SC | 6.5 | 11 | - | 6 | 15 | - | 6 | 16.5 | - |
| C+ | 14 | 16.5 | - | 15.5 | 10 | - | 13.5 | 16.5 | - |
| IG | 6.5 | 2 | 6 | 5 | 5.5 | 8 | 3 | 3 | 2 |
| EG | 1 | 1 | 7 | 1.5 | 1 | 6 | 15.5 | 7 | 12 |
| DL | 11.5 | 4 | 4.5 | 10 | 7.5 | 10 | 4.5 | 2 | 1 |
| DLS | 3 | 6.5 | 8.5 | 3 | 7.5 | 5 | 7 | 6 | 8 |
| LRP | 16.5 | 13 | 13.5 | 14 | 14 | 4 | 2 | 4 | 10.5 |
| RA | 8 | 11 | 10 | 4 | 3.5 | 3 | 8 | 12.5 | 10.5 |
| RoA | 4 | 6.5 | 11.5 | 10 | 3.5 | 2 | 15.5 | 11 | 14 |
| LA | 2 | 8 | 13.5 | 7.5 | 5.5 | 1 | 11.5 | 15 | 13 |

Per modality: ■ Top 1 □ Top 2-4 ▨ Bottom 2-4 ■ Bottom 1

Table 12: Average ranking of the Transformer architectures. Coloring coincides with top and bottom positions.

Table 11 shows the average metric rank for CNN architectures while Table 12 shows the average metric rank based on Transformer architectures. Attention methods can only be applied to Transformer architectures. We observe only minor differences between both tables.

### K.4 Differences in ranking order between model architectures

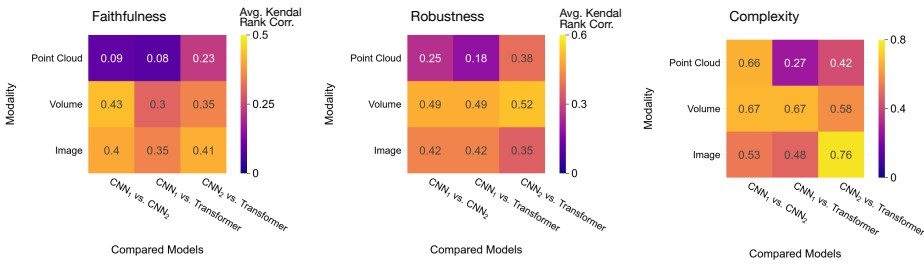

**a.** Average Rank Correlation of Evaluation Metrics between Models

Figure 11: Kendall's-$\tau$ rank correlation between model architectures averaged over datasets and faithfulness criteria.

We compare the difference in faithfulness rankings of attribution methods between CNN and Transformer architectures, as biased methods should be less faithful to the model. To this end, we compute the Kendals-$\tau$ rank correlation between each of the three architectures per dataset and compute their average correlation per modality (see Figure 11). We observe a positive correlation between all rankings. For the point cloud modality, however, the correlation is significantly lower than for the other two modalities, indicating less similar rankings between model architectures. For volume and image modality, the similarity between CNN architectures is generally higher.

## L   Shortcomings of evaluation metrics in practice

While all metrics are theoretically very well founded, we observed for some metrics shortcomings in applications:

Casting saliency maps from 64-bit to 32 or 16-bit to save disk space in such large evaluations is not recommended, as our experiments showed that even 32-bit precision can lead to numerical instability, resulting in all-zero saliency maps and *nan* or *inf* evaluation scores.

Sufficiency evaluates the likelihood that observations with the same saliency maps also share the same prediction label. In practice, this requires several saliency maps from observation with the same prediction label. While this works well on datasets with a small number of labels and balanced sampling, for datasets like IMN with 1000 labels, the probability is almost zero that at least 5-10 sampled observations in a set of sizes 50 or 100 have the same label.

Sequence outputting metrics that alter the input space, such as PF, RP, or ROAD, are only limited suitable for binary prediction tasks. When the input object is too noisy/perturbed to predict accurately, the probability for each class is 0.5 resulting in sequences converging against 0.5 and not 0. While the resulting AUC (or AOC in the case of RP) can be compared between XAI methods within this task, between tasks the AUC would be biased as the area for the binary task would always be larger.

ROAD scores are arrays of binary sequences which are averaged to one sequence. The amount of noise has to be carefully tuned (also depending on the underlying model) as otherwise, all binary sequences in the array are only 0 or 1.

LLE approximates the Lipschitz smoothness through several forward passes of a batch of observations. In application, this results in a large amount of RAM used (depending on modality) if the approximation should be stable. While the computation is relatively fast on a GPU, stable approximations exceed 40GB of VRAM by far and have to be partitioned. For the Transformer architectures, computation on the CPU for our amount of data was too slow to be feasible.

Effective complexity uses a nominal threshold value to determine attributed features. Even through normalization of the saliency maps, the threshold value can have a large effect on the results, differing between observations, and we would suggest tuning it per dataset.

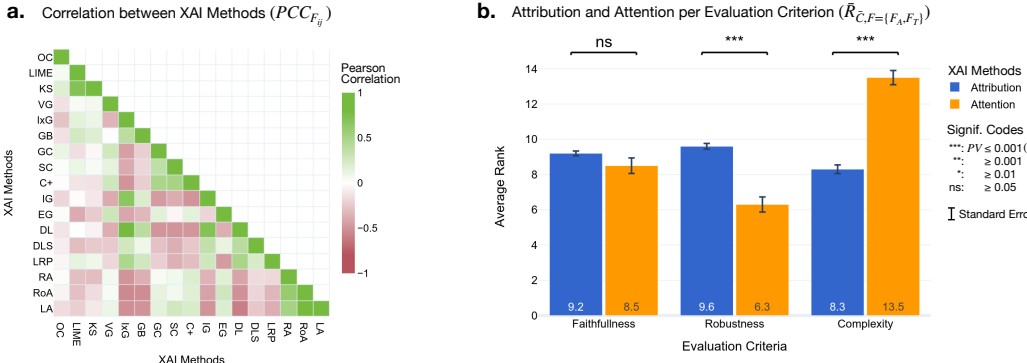

Figure 12: **a.** Correlation in ranking between XAI methods (see Equation 8). **b.** Average ranks of attribution (across all architectures, but across Transformers-only is very alike) and attention (Transformers-only) methods. The standard error of the mean is larger for attention methods.

IROF superpixel segmentation can result in very defined or binary structures such as in the AMN dataset in only two superpixels (object and background), ignoring finer structures.

As elaborated, all complexity metrics flatten the input object treating it as a vector and ignoring spatial dependencies.

## M    What behavioral similarities exist among XAI methods?

To resolve inconsistencies in current research for method selection, our analysis of XAI behavior focuses on two key aspects: similarities among methods and distinct performance trends. Similarity is important in method selection because choosing a heterogeneous set of XAI methods includes different perspectives on the explanation, which is often advantageous in application. Specifically, we analyze the similarity between single methods and the subgroups of attention and attribution methods, obtaining findings 1-4, answering our main question. Figure 12 (a.) shows the correlation in ranking between XAI methods, indicating their relative similarity, based on Equation 8.

We observe that methods belonging to methodological similar groups are positively correlated: Linear surrogate methods (LIME, KS), CAM methods (GC, SC, C+), and attention methods (RA, RoA, LA). Also, CAM and attention methods are slightly positively correlated, indicating their similar attributing to local regions. We would advise not restricting access to such methodological subgroups to preserve method diversity in application.

Contrarily to other method subgroups, the Shapely value approximating SHAP methods (EG, KS, and DLS) are not correlated. Also, their performances in Table 2 differ extensively. This observation is consistent with the results of Molnar et al. [41], which are, however, not in the context of XAI evaluation. Therefore, it is advisable not to select a single SHAP method with the expectation of achieving similar results to others but rather to employ multiple such methods.

CAM and attention methods negatively correlate with IxG, GB, IG, and DL, which contrarily attribute to single pixels, resulting in more fine-grade saliency maps. Interestingly, we observe a very strong positive correlation between IG/IxG, DL/IxG, and IG/DL, indicating very homogeneous behavior between the methods, even though they are based on different mathematical mechanisms. We would strongly recommend mixing such single-pixel and local-region attributing methods, not only for the diversity in visualization but also because of their different performance in evaluation.

Due to the success of Transformers, attention methods are one of the most emerging subgroups of XAI methods. This raises a pressing question for users: should they exclusively use Transformer-based models for attention methods, or can architecture-independent attribution methods still provide equal or superior explanations? When comparing the average ranking between both groups for all criteria, we observe in Figure 12 (b.) a large difference in complexity and a smaller difference in robustness while the difference in faithfulness is insignificant. The comparatively high robustness of attention methods extends across all methods and modalities, as can be seen from Table 2. However, attention

methods exhibit a substantially higher SD between faithfulness metrics compared to attribution methods (see Table 2), rendering the faithfulness results for attention methods more uncertain. Considering our concerns about the complexity metrics as well as the high SD between faithfulness metrics, we would subsequently advocate only for prioritizing attention methods over attribution methods if robustness is the most desired criterion.

# N    Sensitivity of the XAI methods hyperparameter

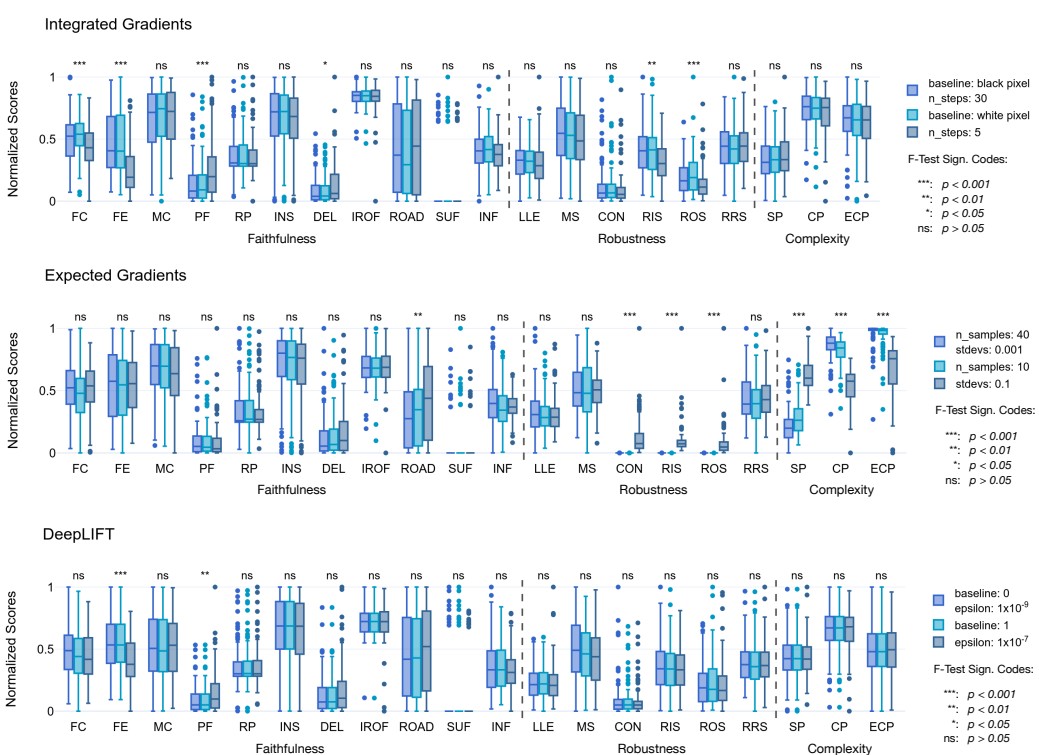

Figure 13: Metric score distributions for the top five ranked XAI methods with hyperparameters, evaluated on the image modality. Results are shown for three realistic hyperparameter combinations.

We conducted an ablation study on the top three performing XAI methods with hyperparameters for the imaging modality to assess the robustness of their performance. Figure 13 presents the evaluation results for three different hyperparameter combinations applied to EG, IG, and DL on ImageNet. An F-Test was used to determine whether variations in hyperparameters resulted in significant differences in the evaluation metrics. Most tests did not indicate significant changes; however, the 'stdevs' parameter in EG, which introduces noise to subsampled images similar to SmoothGrad, significantly reduced complexity while enhancing robustness scores.

# O   Effect of Smooth- and VarGrad

**Faithfulness**

|  |  | $\hat{\mu}$ | $x_{n/2}$ | $\hat{\sigma}$ | FC | FE | MC | PF | RP | INS | DEL | IROF | ROAD | INF |
|---|---|---|---|---|---|---|---|---|---|---|---|---|---|---|
| IxG | Normal | 0.391 | 0.413 | 0.214 | 0.562 | 0.350 | 0.247 | 0.095 | 0.748 | 0.281 | 0.093 | 0.544 | 0.477 | 0.514 |
|  | SmoothGrad | 0.398 | 0.392 | 0.238 | 0.596 | 0.640 | 0.263 | 0.065 | 0.763 | 0.275 | 0.108 | 0.515 | 0.243 | 0.509 |
|  | VarGrad | 0.356 | 0.365 | 0.218 | 0.465 | 0.434 | 0.255 | 0.047 | 0.773 | 0.230 | 0.077 | 0.549 | 0.368 | 0.362 |
| IG | Normal | 0.390 | 0.411 | 0.175 | 0.462 | 0.437 | 0.514 | 0.092 | 0.670 | 0.313 | 0.135 | 0.507 | 0.385 | 0.384 |
|  | SmoothGrad | 0.394 | 0.385 | 0.210 | 0.525 | 0.523 | 0.362 | 0.073 | 0.811 | 0.275 | 0.139 | 0.470 | 0.352 | 0.408 |
|  | VarGrad | 0.384 | 0.393 | 0.221 | 0.468 | 0.563 | 0.316 | 0.090 | 0.722 | 0.203 | 0.064 | 0.517 | 0.318 | 0.582 |
| EG | Normal | 0.392 | 0.397 | 0.223 | 0.482 | 0.541 | 0.293 | 0.070 | 0.719 | 0.219 | 0.105 | 0.574 | 0.312 | 0.608 |
|  | SmoothGrad | 0.383 | 0.368 | 0.236 | 0.469 | 0.520 | 0.215 | 0.085 | 0.776 | 0.244 | 0.083 | 0.540 | 0.268 | 0.626 |
|  | VarGrad | 0.407 | 0.421 | 0.200 | 0.446 | 0.518 | 0.394 | 0.095 | 0.671 | 0.239 | 0.128 | 0.639 | 0.396 | 0.546 |
| DL | Normal | 0.422 | 0.426 | 0.241 | 0.464 | 0.623 | 0.350 | 0.069 | 0.846 | 0.253 | 0.112 | 0.491 | 0.389 | 0.618 |
|  | SmoothGrad | 0.345 | 0.349 | 0.223 | 0.461 | 0.561 | 0.327 | 0.120 | 0.729 | 0.222 | 0.063 | 0.509 | 0.371 | 0.085 |
|  | VarGrad | 0.348 | 0.323 | 0.226 | 0.517 | 0.543 | 0.319 | 0.093 | 0.734 | 0.204 | 0.100 | 0.537 | 0.328 | 0.107 |
| DLS | Normal | 0.353 | 0.378 | 0.244 | 0.529 | 0.514 | 0.321 | 0.073 | 0.765 | 0.198 | 0.079 | 0.554 | 0.435 | 0.063 |
|  | SmoothGrad | 0.362 | 0.380 | 0.217 | 0.485 | 0.538 | 0.397 | 0.109 | 0.668 | 0.232 | 0.122 | 0.619 | 0.363 | 0.085 |
|  | VarGrad | 0.377 | 0.360 | 0.261 | 0.576 | 0.657 | 0.383 | 0.102 | 0.849 | 0.245 | 0.068 | 0.459 | 0.336 | 0.100 |

**Robustness**

|  |  | $\hat{\mu}$ | $x_{n/2}$ | $\hat{\sigma}$ | LLE | MS | RIS | ROS | RRS |
|---|---|---|---|---|---|---|---|---|---|
| IxG | Normal | 0.407 | 0.467 | 0.422 | 0.467 | 1.000 | 2E-06 | 0E+00 | 0.568 |
|  | SmoothGrad | 0.368 | 0.378 | 0.412 | 0.462 | 1.000 | 2E-06 | 0E+00 | 0.378 |
|  | VarGrad | 0.344 | 0.345 | 0.405 | 0.345 | 0.991 | 6E-04 | 1E-03 | 0.384 |
| IG | Normal | 0.361 | 0.317 | 0.414 | 0.317 | 1.000 | 1E-06 | 0E+00 | 0.488 |
|  | SmoothGrad | 0.352 | 0.320 | 0.411 | 0.320 | 1.000 | 0E+00 | 0E+00 | 0.439 |
|  | VarGrad | 0.370 | 0.425 | 0.411 | 0.425 | 1.000 | 1E-06 | 0E+00 | 0.427 |
| EG | Normal | 0.366 | 0.371 | 0.412 | 0.460 | 1.000 | 1E-06 | 0E+00 | 0.371 |
|  | SmoothGrad | 0.372 | 0.365 | 0.410 | 0.502 | 0.988 | 1E-03 | 2E-03 | 0.365 |
|  | VarGrad | 0.398 | 0.469 | 0.418 | 0.520 | 1.000 | 0E+00 | 0E+00 | 0.469 |
| DL | Normal | 0.369 | 0.399 | 0.412 | 0.448 | 1.000 | 1E-06 | 2E-06 | 0.399 |
|  | SmoothGrad | 0.311 | 0.073 | 0.430 | 0.073 | 0.992 | 5E-03 | 5E-04 | 0.485 |
|  | VarGrad | 0.300 | 0.087 | 0.426 | 0.087 | 1.000 | 3E-03 | 1E-04 | 0.409 |
| DLS | Normal | 0.315 | 0.126 | 0.367 | 0.029 | 0.918 | 1E-01 | 9E-02 | 0.406 |
|  | SmoothGrad | 0.320 | 0.099 | 0.429 | 0.099 | 0.995 | 5E-03 | 3E-03 | 0.496 |
|  | VarGrad | 0.302 | 0.093 | 0.424 | 0.093 | 0.996 | 4E-04 | 1E-03 | 0.420 |

**Complexity**

|  |  | $\hat{\mu}$ | $x_{n/2}$ | $\hat{\sigma}$ | SP | CP | ECP |
|---|---|---|---|---|---|---|---|
| IxG | Normal | 0.419 | 0.376 | 0.176 | 0.268 | 0.612 | 0.376 |
|  | SmoothGrad | 0.427 | 0.391 | 0.165 | 0.283 | 0.607 | 0.391 |
|  | VarGrad | 0.940 | 0.959 | 0.062 | 0.871 | 0.959 | 0.990 |
| IG | Normal | 0.360 | 0.289 | 0.163 | 0.246 | 0.546 | 0.289 |
|  | SmoothGrad | 0.902 | 0.883 | 0.038 | 0.876 | 0.945 | 0.883 |
|  | VarGrad | 0.888 | 0.880 | 0.061 | 0.880 | 0.952 | 0.832 |
| EG | Normal | 0.853 | 0.846 | 0.080 | 0.846 | 0.936 | 0.777 |
|  | SmoothGrad | 0.932 | 0.951 | 0.067 | 0.858 | 0.951 | 0.987 |
|  | VarGrad | 0.838 | 0.845 | 0.096 | 0.845 | 0.930 | 0.738 |
| DL | Normal | 0.840 | 0.816 | 0.062 | 0.793 | 0.909 | 0.816 |
|  | SmoothGrad | 0.213 | 0.175 | 0.191 | 0.175 | 0.419 | 0.044 |
|  | VarGrad | 0.196 | 0.157 | 0.175 | 0.157 | 0.387 | 0.043 |
| DLS | Normal | 0.188 | 0.182 | 0.131 | 0.182 | 0.322 | 0.059 |
|  | SmoothGrad | 0.229 | 0.189 | 0.216 | 0.189 | 0.462 | 0.035 |
|  | VarGrad | 0.165 | 0.137 | 0.137 | 0.137 | 0.314 | 0.043 |

Table 13: Mean evaluation metric scores for the top 5 XAI methods when applied normal, with SmoothGrad and with VarGrad. The results are for the image modality.

Table 13 shows the mean evaluation metric score of the top five ranked methods when computing the saliency map either normally, with SmoothGrad, or with VarGrad. As in Table 2 we show the mean $\hat{\mu}$, median $x_{n/2}$, and SD $\hat{\sigma}$ for each row. The results are for the image modality. We observe no clear trend if both advancements improve one XAI method. Only in terms of complexity, SmoothGrad improves three out of five methods substantially. Due to the noisy sampling of the saliency maps by SmoothGrad, we assume that the resulting saliency maps are more localized, thus reducing complexity.

# P  Wilcoxon-Mann-Whitney test between all rankings

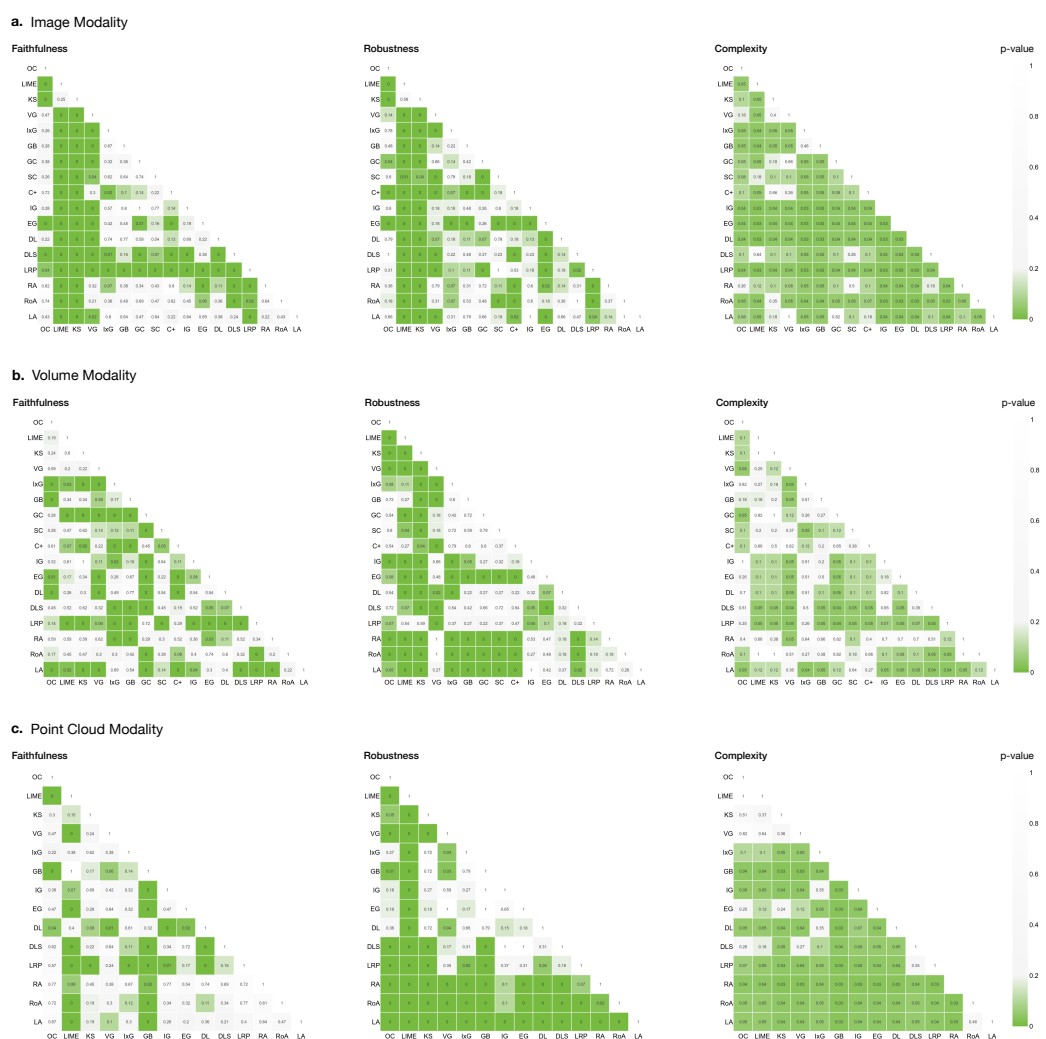

Figure 14: P-value of the Wilcoxon-Mann-Whitney test between all rankings in Table 2 for each modality and criteria. Through the p-value matrices, we can determine which XAI methods are significantly differently ranked or could be interpreted as a tie position. We would advise however to also take the other results in Table 2 into account as the power of the test is limited.

