# OpenReview forum: "Navigating the Maze of Explainable AI: A Systematic Approach to Evaluating Methods and Metrics"
_NeurIPS.cc/2024/Datasets_and_Benchmarks_Track — NeurIPS 2024 Track Datasets and Benchmarks Poster_

### Official Review · Reviewer_JMZx · 2024-07-16
**Benchmarking and Assessing XAI Methods in 3D Modality**

**Rating:** 5
**Confidence:** 3
**Correctness:** Yes.
**Clarity:** This paper is clear and well-written.

**Review:**

While the author advocates for an aggregated evaluation of XAI methods, this approach may prove time-intensive and not always yield practical insights. Currently, the work appears more exploratory than providing a reliable benchmark for the XAI community.

A more nuanced analysis could benefit the field by delving into the design choices inherent in XAI methods themselves. It is important to acknowledge that certain evaluation metrics might inherently favor specific methods due to their computational design. For example, varying baseline choices (masking value) in methods like Kernel SHAP or Integrated Gradients (IG) can lead to different outcomes. Investigating the effects of various masking techniques could provide deeper insights into each method’s strengths and weaknesses. This exploration is crucial as aggregating rankings across different evaluation tasks might introduce biases if the chosen methods themselves are biased.

Moreover, some of the results in the paper are as anticipated. For example, it reports no significant differences across architectures. This finding isn't surprising since many XAI methods are designed to be model-agnostic.

Another concern arises with the performance disparities among SHAP method subgroups. In theory, the Shapley value should converge regardless of the imputation method used. This discrepancy warrants a deeper investigation into why SHAP methods vary so significantly in performance.

**Strengths:**

The author offers a comprehensive examination of XAI methods across various image datasets, providing guidance for selecting XAI techniques suitable for high-dimensional inputs, such as 3D data.

**Additional Feedback:**

None

**Documentation:**

Yes

**Limitations:**

Yes.

**Opportunities For Improvement:**

Rather than attempting an exhaustive benchmark of every XAI method, it would be more pragmatic to selectively focus on those methods that are best suited for 3D modalities. Additionally, a detailed analysis explaining why these biases occur and proposing strategies to mitigate them would greatly benefit the research community. Moreover, beyond simply examining existing XAI evaluation tasks, it would be more valuable if the author could develop a new benchmark or evaluation framework specifically tailored for 3D modalities. This would provide more relevant insights.

**Relation To Prior Work:**

Yes, in Section 5.

**Summary And Contributions:**

This study provides a thorough evaluation of existing XAI methods across 3D modalities and various model architectures. It demonstrates that rankings can differ significantly based on the evaluation metrics used. Additionally, the author proposes the analysis of rank statistics as a method for comparing XAI techniques.

---

> ### Author Rebuttal · Authors · 2024-08-16
>
> We thank reviewer JMZx for their feedback. The points raised regarding the focus on 3D modalities and methods as well as the sensitivity of hyperparameters are addressed in our response to all reviewers above. All other concerns are addressed point-by-point below.
>
> **W1** ("Rather than attempting an exhaustive benchmark [...]", "Moreover, beyond simply examining existing XAI evaluation tasks [...]")**:** Please see the answer in Concern 1 of our response to all reviewers above.
>
> **W2** ("For example, varying baseline choices (masking value) [...]", "Additionally, a detailed analysis explaining why these biases [...]")**:** If we understand this point correctly, it regards biases arising due to hyperparameters of the XAI methods. In the general response (concern 2) we discuss the sensitivity of our ranking across multiple hyperparameter settings by means of an additional ablation.
>
> **W3** ("Moreover, some of the results in the paper are as anticipated. For example [...]")**:** There appears to be a misconception regarding the term "model agnostic" in the context of XAI. In XAI, "model agnostic" typically means that the methods can be applied to any type of machine learning or deep learning model [1, 2], *not* that they will yield the same results regardless of the underlying model. Thus, there is no reason to anticipate consistent results across all model architectures. Demonstrating that only little variability exists for both model-agnostic and model-specific methods on such a large benchmark is a significant new finding for the community.
>
> Thank you again for the feedback. As we believe to have resolved your comments, please let us know if there are any remaining concerns that would hinder a recommendation for acceptance.
>
> ---
>
> ​​[1] Christoph, M. (2020). Interpretable machine learning: A guide for making black box models explainable. Leanpub.
>
> [2] Ribeiro, M. T., Singh, S., & Guestrin, C. (2016, August). " Why should i trust you?" Explaining the predictions of any classifier. In Proceedings of the 22nd ACM SIGKDD international conference on knowledge discovery and data mining (pp. 1135-1144).

---

> > ### Comment · Reviewer_JMZx · 2024-08-21
> >
> > Thanks for the clarification. I've adjusted my score.
> >
> > However, I believe further experiments of hyperparameters for each method are necessary. For example, hyperparameters for removal-based approaches, such as LIME, the Shapley value family, and Occlusion are missing. Additionally, I would appreciate insights into the reasons why metrics might agree or disagree across different methods.

---

> > ### Author Response · Authors · 2024-08-25
> >
> > Dear Reviewer JMZx,
> >
> > Thank you for your thorough review of our response. Regarding your suggestion for further experiments on hyperparameter sensitivity, we would like to note that we have already evaluated two of the three Shapley value methods, specifically DeepLIFT SHAP and Expected Gradients (GradientSHAP), as they are two of the five best-evaluated methods. The current scope of our work includes the sensitivity analysis of the five best-evaluated methods to realistic configurations of their hyperparameter. Analyzing all other methods including all possible hyperparameter combinations and unrealistic configurations we deem however out of scope for this paper as our recommendations to practitioners only focus on the consistently best-performing methods. Additionally, for 5 values per hyperparameter, we estimate in total 2109 GPU-days for computation and evaluation of all hyperparameter combinations with one metric, ignoring even the testing of several metrics (which would scale on average linearly depending on the metric) and very minor hyperparameters as e.g. the learning rate in LIME. Thus we would argue that while these experiments would be of course interesting, our current analysis covers all important methods to validate our claims. We would appreciate further clarification on the necessity of these tests in order to better understand how they contribute to the overall value of the paper for the community.
> >
> > Concerning metric disagreement, we have already addressed this topic in general in Appendix Section J, where we also illustrate why Region Perturbation tends to favor Occlusion. Expanding this analysis to cover all 340 possible combinations would extend beyond the scope of this paper and the NeurIPS Dataset and Benchmark track, as evaluating each combination could constitute a separate study.

---

### Official Review · Reviewer_AiHf · 2024-07-25
**Meta-evaluation of XAI Methods**

**Rating:** 7
**Confidence:** 4
**Correctness:** Yes, the paper is correct.
**Clarity:** The paper is clear.

**Review:**

The paper presents insights into the performance of different XAI methods across various modalities and across different models. It identifies corner cases, such as the inconsistent performance of SHAP. The use of faithfulness, robustness, and complexity as evaluation criteria is interesting.

**Strengths:**

1. The paper presents a comprehensive benchmark for testing XAI methods.
2. The new approach employs diverse model architectures and input modalities.
3. Their evaluation of existing XAI methods has identified inconsistencies in current XAI evaluations.
4. There is a good likelihood that this benchmark will be used by others seeking to evaluate their XAI approaches.

**Additional Feedback:**

None

**Documentation:**

The data set is well documented in my opinion.

**Ethics:**

No concerns.

**Limitations:**

The alignment of evaluation criteria with ground truth and human interpretation could have been better discussed.

**Opportunities For Improvement:**

1. It is unclear how easily this benchmarking approach can be applied to new XAI approaches.
2. What interface should a new XAI developer use to connect to this system/benchmark? Is it easy to do? Some of that discussion would be helpful.
3. Could you or a third-party host this as a service? If so, how would one "upload" a new XAI approach to this service?

If we could document the connection to its use case well, this could have a good impact.

**Relation To Prior Work:**

There is a good connection to related prior work.

**Summary And Contributions:**

This paper introduces a new way to evaluate explainable AI (XAI) methods. It presents a large-scale benchmark called LATEC that tests 17 prominent XAI methods with 20 different metrics. The benchmark also considers various model architectures and input types, which helps understand the generalization of XAI performance. The results also highlight the inconsistencies in current evaluations of XAI methods.

---

> ### Author Rebuttal · Authors · 2024-08-16
>
> We thank reviewer AiHf for the elaborate and helpful feedback. All weaknesses are addressed point-by-point below.
>
> **W1:** Thank you for pointing this out. Based on your feedback, we have updated the README of our repository. It now provides clearer instructions on how to add new XAI methods or metrics to the benchmark in just three simple steps. See the [run your own experiments](https://github.com/kjdhfg/LATEC?tab=readme-ov-file#run-your-own-experiments) section in the README.
>
> **W2:** The benchmark is easily installable via pip and can be controlled through the command line. Based on your feedback, we have updated the README with a step-by-step guide for the requirements and installation process. All saliency map and evaluation score datasets can be downloaded directly from the internet without needing to log into a website. The computer vision datasets used in the experiments will automatically download and extract when an experiment is initiated. However, users must manually download the CoMA and RESISC45 datasets after creating a free account on the respective websites.
>
> **W3:** Thank you for this idea. We agree that implementing a web app for ranking final scores would be valuable, allowing users to compare their results similar to traditional deep learning benchmarks. If we are able to secure funding to host such a web service, this would be a promising extension to our project.
>
> Thank you once more for your constructive feedback. As we believe to have resolved your comments, please let us know in case you have remaining suggestions to further increase the quality of our work.

---

### Official Review · Reviewer_xgqg · 2024-07-25
**paper review**

**Rating:** 6
**Confidence:** 3
**Correctness:** Yes
**Clarity:** Yes

**Review:**

This paper provides a comprehensive evaluation framework for existing XAI methods from many perspectives including diverse datasets, methods, and evaluation metrics. The evaluation process is solid and covers most of the existing XAI methods. The investigation into the misleading of some evaluation metrics is interesting and important for evaluating XAI approaches. This paper provides some valuable insights in how to evaluate the XAI method in a more robust way.

**Strengths:**

1.	The designed framework provides a comprehensive evaluation tool for existing XAI methods and has the potential to formalize the research area.
2.	The writing is clear and easy to follow. Some important insights in section 3 about the potentially misleading evaluation metrics and how to proceed for a robust evaluation are interesting and conveyed in a clear way.
3.	The experiments are comprehensive.

**Additional Feedback:**

N/A

**Documentation:**

Yes

**Ethics:**

No.

**Limitations:**

Yes

**Opportunities For Improvement:**

1.	It would be good if the authors could discuss more about the performance difference on different kinds of datasets (e.g., image vs point cloud).
2.	It would also be interesting if the authors could discuss more about the potential correlation of different evaluation metrics.

**Relation To Prior Work:**

Yes

**Summary And Contributions:**

This paper introduced a comprehensive benchmark analysis of 17 existing XAI methods. Some key shortcomings of existing XAI benchmarks, such as small datasets, lack of attention in model architecture, or incomplete or misusage of evaluation metrics, have been discussed in the paper. To evaluate the performance with robust results, the authors utilized 20 distinct metrics and diversified datasets, including images, 3D point clouds, and volume data. This paper also verified the performance from three perspectives: faithfulness, robustness, and complexity. They open-sourced their code and evaluation process.

---

> ### Author Rebuttal · Authors · 2024-08-16
>
> We appreciate the reviewer xgqg’s insightful feedback and have addressed all raised weaknesses point-by-point below.
>
> **W1:** In Section 4, we discuss the dependence of XAI method performance on datasets (Line 237) and modalities (Line 254). However, our initial analysis focused mainly on extreme cases like the linear surrogate and CAM methods. We agree that a broader discussion is needed. To address this, we expanded the paragraph in Line 265 to emphasize that results across modalities are generally more consistent in terms of robustness but show greater variation in faithfulness and complexity. Additionally, the standard deviation of robustness metrics between volume and point cloud data suggests that modality can influence disagreements between metric perspectives.
>
> **W2:** Thank you for pointing this out. In Appendix J, Figure 8, we present and discuss the distance matrices between metric rankings across all image datasets, highlighting metrics that frequently agree or disagree. However, we agree that adding correlation analysis can offer additional insights. While Euclidean distance and Pearson correlation both measure similarity, correlation focuses on trends, whereas distance measures actual differences. Therefore, we computed the correlation matrix for each evaluation criterion and modality. The new results for faithfulness metrics on the imaging modality are shown in Figure (c) of the uploaded PDF, with a detailed discussion in Appendix J. For example, in our assessment of faithfulness, we found that metrics involving incremental pixel perturbation (PF, INS, DEL) and those correlating attribution values with predicted logits (FC, FE) are positively correlated. However, metrics specifically designed to address limitations in these methods, such as ROAD (which addresses the out-of-distribution issue in pixel perturbation) and MC (which incorporates uncertainty rather than logits), are interestingly even negatively correlated with them. Regarding robustness, the relative stability metrics show a positive correlation.
>
> Thank you once more for your constructive feedback. As we believe to have resolved your comments, please let us know if there are any remaining concerns that would prevent you from increasing your score.

---

### Official Review · Reviewer_6Gad · 2024-07-29
**A carefully constructed large scale XAI benchmark**

**Rating:** 7
**Confidence:** 5
**Correctness:** the claims are correct and sound
**Clarity:** the paper is clear and well written

**Review:**

Overall, this is a carefully constructed benchmark which is larger scale than previous benchmarks. The paper provides a few reasonable and interesting insights into XAI method performance and evaluation metric robustness as well as offers some advice for practitioners. However, as the paper's main motivation is that the problem of metric disagreement is a major and pronounced one, the paper would benefit from showing stronger evidence of frequent metric disagreement that goes beyond Figure 2 as Figure 2 showcases arguably minor disagreements between metrics and such behavior may have reasonable potential explanations that do not imply flaws in traditional XAI evaluations. The paper also finds that metric variance is stable across architectural design properties, which highlights that existing metrics are mostly well designed. Additionally, I would recommend including a detailed comparison with other existing comprehensive XAI benchmarks (e.g., [1,2]) which is currently overlooked. Overall, the differentiating contributions of the LATEC benchmark are (1) somewhat larger scale compared to previous benchmarks including more metrics, (2) implementation of best practices for careful robust metric result aggregation, and (3) including additional modalities such as 3D point clouds and volumes. I would say the community would benefit from acceptance of this paper, but the acceptance decision should be considered within the context of other papers and the high bar of NeurIPS Benchmark track as the impact of the current submission is potentially limited.

References
[1] Zhang, Y., Gu, S., Song, J., Pan, B. and Zhao, L., 2023. XAI Benchmark for Visual Explanation. arXiv preprint arXiv:2310.08537.

[2] Li, X., Du, M., Chen, J., Chai, Y., Lakkaraju, H. and Xiong, H., 2023, December. M4: A Unified XAI Benchmark for Faithfulness Evaluation of Feature Attribution Methods across Metrics, Modalities and Models. In NeurIPS.

**Strengths:**

* Large scale evaluation: 17 methods, large set of model design choices, 20 metrics. This study provides a larger scale evaluation than previous benchmarks.
* This work goes beyond evaluation for 2D images and also includes 3D point clouds and volumes
* Systematic evaluation of metrics themselves and evaluation of their ranking behaviors — this is helpful
* It is interesting that the metric variance is stable across architectural design properties, which is a desirable property of metrics. This is a valuable point that the study highlights even if it does not indicate a problem with metrics.
* A set of useful takeaways and practical recommendations based on the results of the benchmark

**Additional Feedback:**

please, see the review and opportunities for improvement sections

**Documentation:**

datasets are documented well

**Ethics:**

no ethics concerns

**Limitations:**

limitations are addressed

**Opportunities For Improvement:**

* Differences from other existing benchmarks for XAI such as [1,2] are not discussed, [1] is not even cited
* “For comparison reasons, we only consider the original methods without adaptations, as other works already showed that advancing methods by VarGrad or SmoothGrad can, in general, improve results.” — if you are claiming that your benchmark enables practitioners to choose *the best* method, it would be nice to include those improved methods with modifications too to see where they stand.
* “We qualitatively tuned the XAI hyperparameters” -- if one sets out to solve the problem of metric inconsistency and lack of evaluation robustness, one should either systematically tune hyperparameters or show ablations/sensitivity analysis that changing hyperparameters does not lead to drastic changes in method ranking. Hyperparameters could be tuned using frameworks such as Optuna under quantitative objectives or, if there is a goal to avoid that for the lack of a reasonable single objective, even hyper parameter tuning under qualitative objectives could be automated and scaled up through the use of vision-language models such as GPT-4o or Claude 3 models (or LLAVA for the open-source options) in the “LLM-as-a-judge” setting.
* Figure 2a shows that the metrics disagree on ranking faithfulness of GC vs IG, but otherwise it is actually hard to say that the ranking completely lacks robustness. Another way to explain such behavior is that the methods are potentially on par and some metrics may emphasize slightly different aspects of faithfulness. It would be nice to show stronger evidence of frequent metric disagreement.
* Median-aggregate-then-rank is a good approach to aggregate metrics, and a good design practice for a benchmark. However, then average ranks are computed and average rank can also be skewed by outliers (e.g. lower quality metrics), it may be helpful to aggregate by median ranks. Would be nice to show median ranks next to average ranks.
* “We believe that understanding why metrics disagree and the situations in which this occurs is vital for evaluating XAI” — does the paper provide any insight into why?
* Would be nice to have a section in the main body on what each evaluation metric emphasizes to give practitioners a broad clear picture .
* When computing ranking, it would be helpful to consider statistical significance of performance differences (e.g. through Wilcoxon rank-sum test) and allow ties in rank computation like for example is done in tabular deep learning literature [3]. It would be helpful to recompute the benchmark ranking results with the ties.

With these weaknesses addressed, I am open to increasing my score.

References

[1] Zhang, Y., Gu, S., Song, J., Pan, B. and Zhao, L., 2023. XAI Benchmark for Visual Explanation. arXiv preprint arXiv:2310.08537.

[2] Li, X., Du, M., Chen, J., Chai, Y., Lakkaraju, H. and Xiong, H., 2023, December. M4: A Unified XAI Benchmark for Faithfulness Evaluation of Feature Attribution Methods across Metrics, Modalities and Models. In NeurIPS.

[3] Levin, R., Cherepanova, V., Schwarzschild, A., Bansal, A., Bruss, C.B., Goldstein, T., Wilson, A.G. and Goldblum, M., 2022. Transfer learning with deep tabular models. arXiv preprint arXiv:2206.15306.

**Relation To Prior Work:**

relation to prior XAI benchmarking work is overlooked and should be discussed in more detail

**Summary And Contributions:**

The paper highlights and addresses the important problem of metric inconsistency when evaluating XAI methods. The paper proposes a large-scale benchmark LATEC and systematically evaluates 17 popular XAI methods across 20 metrics while varying model architectures and input modalities. The comprehensive benchmark provides insight to practitioners for method selection and the paper also finds that a consistently well-performing method Expected Gradients is overlooked in the XAI literature. The LATEC benchmark design choices follow best practices of robust result aggregation.

---

> ### Author Rebuttal · Authors · 2024-08-16
>
> Thank you for the very detailed and thoughtful feedback, and for taking the time to read our general reply, as well as considering our point-by-point comments here:
>
> **W1:** Thank you for pointing out these studies. We incorporate [1] by adding it to Tab. 1 and compare to them in Line 305. Our findings show alignment with [1, 2] for RA, GC, IxG, and VG, and, again, variation in results for the faithfulness evaluation of IG. We revised the paragraph at Line 36 to emphasize the methodological differences between our work and existing benchmarks, particularly their reliance on 2D image data, focus on faithfulness, and inconsistent small subsets of XAI methods and metrics. We emphasize that the lack of metric validation in other benchmarks undermines the reliability of their rankings.
>
> **W2:** We agree that evaluation results including SmoothGrad and VarGrad would improve the manuscript. In response, we conduct an ablation study focused on the top 5 XAI methods for image data, as both extensions significantly increase computation time. Contrary to [4], we find no substantial improvements w.r.t faithfulness or robustness (see Tab. e in the uploaded PDF), but SmoothGrad notably reduces complexity by producing more localized saliency maps. These findings are discussed in the new App. O, and Sec. 3.3 is updated to note that improvements are seen only in complexity.
>
> **W3:** We agree and have conducted the sensitivity analysis, as detailed in Concern 2 of our response to all reviewers. Below, we also wish to address the other two points raised:
> We currently qualitatively tune the XAI methods' hyperparameters on 10 observations per dataset, focusing on localization and dataset characteristics. This approach is chosen because qualitative tuning is common in practice, avoids biasing the whole benchmark from optimizing for a single metric, and prevents the prohibitively high computational costs of extensive hyperparameter sweeps, nearly infeasible for a benchmark of this scale.
> We also tuned hyperparameters using vision language models. While LLaVA 1.5 16B and 3D data results were unpromising, ChatGPT 4o provided useful but mostly trivial suggestions based on data characteristics rather than saliency map features. We find this approach interesting for future work and have included it in App. N.
>
> **W4:** It seems there is a misunderstanding, we fully agree with your point. In Line 46, we state that "These metrics reflect the diversity of perspectives on the criterion," emphasizing that perspective disagreement offers key insights into a method's performance. To clarify, we added in Line 122 that selection bias can arise from overfitting to a limited set of perspectives, as different metrics highlight various aspects of the criterion.
>
> Notably, Fig. 2a is meant as an illustrative example of metric disagreement. Extensive quantitative evidence of disagreements is provided by the ranking distance calculations in Fig. 2b, App. I, and the large-scale statistical tests in Fig. 2c.
>
> **W5:** We agree and added the median to all tables in Tab. 1 (see Tab. b) and the new Tab. e in the uploaded PDF. Although the mean and median often coincide, the median is a robust second indicator to the mean when comparing or selecting a method.
>
> **W6:** While our work is the first large-scale quantitative study that uncovers metric disagreement and provides answers to how to address these discrepancies, we acknowledge that the question "why do metrics disagree?" is not the focus. Our study rather serves as a basis for this question, empirically showing for the first time that metric disagreement is dependent on the XAI method. Thoroughly examining and mathematically proving this for all 340 combinations of methods and metrics is beyond the scope of this paper. In response, we revised the statement in Line 191 and highlighted the importance of investigating why metrics disagree as a key area for future work in Line 334.
>
> **W7:** We agree and provide now an abstract overview of the metrics and their functionalities by grouping them into categories: Faithfulness: Correlation in attribution and prediction, feature perturbation, feature removal, feature insertion, and saliency map similarity. Robustness: Local attribution change, relative attribution change, and largest difference in saliency maps. Complexity: Entropy or Gini-Index-based. We have incorporated this classification in Line 100 and refer to App. D for a detailed description of each metric.
>
> **W8:** We agree and have conducted the Wilcoxon-Mann-Whitney test on all XAI method rankings, with results for the image modality in Fig. a of the uploaded PDF. The analysis shows significant differences in complexity rankings, while intermediate methods are tied in faithfulness and robustness. These findings are now discussed in App. P and referenced in Line 206.
>
> Thank you again for your constructive feedback. As we believe to have resolved your comments, please let us know if any remaining concerns would hinder a recommendation for acceptance.
>
> [4] Hooker et al. A Benchmark for Interpretability Methods in Deep Neural Networks. NeurIPS 2019

---

> > ### Comment · Reviewer_6Gad · 2024-08-21
> >
> > Dear Authors,
> >
> > Thank you for your detailed response, adjustments to the paper and clarifications. I am excited to support acceptance of the paper and given the quality of the rebuttal I am happy to raise my score to 7.

---

### Author Rebuttal · Authors · 2024-08-16

We sincerely thank all reviewers for their valuable comments. We appreciate the general consensus among the reviewers regarding the significance of our work for the community: “[...] this benchmark will be used by others” (“AiHf”), “[...] investigation into the misleading of some evaluation metrics is interesting and important for evaluating XAI approaches” (“xgqg”), “I would say the community would benefit from acceptance of this paper [...]” (“6Gad”).

All reviewers agreed on the completeness of the documentation and reproducibility of the results. We also want to highlight again our [code repository](https://github.com/kjdhfg/LATEC) including the detailed readme, explaining how to implement your own XAI methods or metrics, and leveraging our 326k saliency maps and 378k metric scores in the LATEC dataset for standardized benchmarking.

Next to the point-by-point responses to each reviewer, we will address the two main concerns here:

---

**Concern 1: Exclusive focus on 3D modalities and methods tailored to 3D modalities.** While all reviewers appreciated our novel scope across computer vision modalities (“This work goes beyond evaluation for 2D images and also includes 3D point clouds and volumes”), reviewer “JMZx” recommended to solely focus on 3D modalities and methods (“[...] focus on those methods that are best suited for 3D modalities”, “[...] new benchmark or evaluation framework specifically tailored for 3D modalities”). To our knowledge, *there are no 3D-specific XAI methods or metrics* and it is unclear which of the general methods work best for 3D. Instead, gathering this information is one of the primary motivations for our benchmark. We included the only publicly available 3D-adapted methods, LIME and GradCAM, and adapted all other XAI methods and metrics to 3D data if required. Subsequently, we provide the first large-scale insights into XAI method performance and metric behavior on 3D data. In response to this feedback, we revised the introduction to emphasize this point and extend the discussion of the difference in results between 2D and 3D modalities in Line 265.

---

**Concern 2: Sensitivity of the results to XAI hyperparameters.** Reviewers “6Gad” and “JMZx” raised concerns about the sensitivity of the rankings to the hyperparameters of the XAI methods. We agree that sensitivity to hyperparameters is crucial when creating rankings. To address this, we conducted a new extensive ablation study.

Since a comprehensive sensitivity analysis covering all 22 hyperparameters of the XAI methods and their combinations including the evaluation would be computationally infeasible for us (with a grid search of five steps we estimate in total 2109 GPU-days for computation and evaluation of all XAI methods with one metric, ignoring even the testing of several metrics), we conducted an ablation study focusing on the top 5 performing XAI methods for the imaging modality, validating the robustness of their performance. Exemplary, Figure (d) in the uploaded PDF illustrates the evaluation results for three combinations of hyperparameters for IG and EG on ImageNet. We employed an F-Test to determine whether any changes in hyperparameters lead to significantly different evaluations for each metric. Our results show that a large number of these tests are not significant. However, the “stdevs” parameter in EG, which adds noise to the subsampled images similar to SmoothGrad, significantly reduces complexity and increases robustness scores. Additionally, using a white pixel as the baseline in IG, rather than a black one, reduces faithfulness on the OCT dataset. This outcome is expected, as it prevents the method from attributing to the white retinal layers. This study is added to the Appendix (Section N) and a discussion of the results is provided in Line 99.

---

**All changes to the manuscript:** We addressed all additional concerns in our point-by-point responses, conducted the requested experiments which we include in the Appendix, and revised the main paper while retaining the overall structures. Based on the reviewers' feedback, we have made the following improvements to the manuscript (location in manuscript in parentheses):

1. Performed an ablation study on the sensitivity of the XAI methods hyperparameter (new Appendix section N, Line 99 and Figure (d) in the uploaded PDF).
2. Conducted an ablation study on Smooth- and VarGrad (new Appendix section O, Subsection 3.3 and Figure (e)).
3. Conducted the Wilcoxon-Mann-Whitney test between all rankings (new Appendix section P, Figure (a)).
4. Computed and discussed correlation matrices between all metrics (Appendix J and shown in Figure (c)).
5. Added the median to all tables in Table 2 and the new Table (e).
6. Emphasized in the Introduction that this is the first large-scale XAI benchmark for 3D modalities, covering all relevant methods (Section 1).
7. Clarified that metric disagreement is due to differing metric perspectives (Line 122).
8. Stated that our work lays the foundation for future studies on why metrics disagree (Line 334).
9. Described metric subgroups for clarity (Line 100).
10. Expanded discussion on modality differences (Line 265).
11. Added and discussed previously missed related work (Line 305, Table 1).
12. Updated README to guide users on adding methods and metrics, with automatic dataset downloads and precomputed saliency maps (Repository).

---

The uploaded PDF includes the figures and tables added to the manuscript based on the requested experiments. Except for table (b), which is a revision of table 1 in the manuscript, all figures and tables are added to the Appendix. We believe these updates significantly improve the manuscript and resolve the stated concerns of all reviewers.

---

### Decision · Program_Chairs · 2024-09-26

**Decision:**

Accept (Poster)

**Comment:**

Meta review of Navigating the Maze of Explainable AI: A Systematic Approach to Evaluating Methods and Metrics
NeurIPS 2024 Datasets and Benchmarks Track Submission 916

All reviewers agree in their assessment that this paper is a valuable contribution, with appreciation expressed for the scale of the evaluation, the novelty of insights, and the inclusion of 3d data and different architectures. Overall the reviewers agreed that the writing is clear and the paper is practical and useful.

Some reviewers expressed that the paper exhibits some weaknesses, particularly the lack of grounding with existing literature, a comparative lack of depth in the experimental design, and needed guidance for application to new XAI methods. For one reviewer, the authors’ rebuttal and revisions addresses some of these concerns, enough to raise the score.

The majority of reviewers gave final scores above the acceptance threshold (7, 7, and 6) and only a single reviewer gave a score below that threshold (5). Additionally, while that reviewer stated some reservations about the paper (“I would appreciate insights into the reasons why metrics might agree or disagree across different methods.”) the authors provided a reasonable rebuttal (“Expanding this analysis to cover all 340 possible combinations would extend beyond the scope of this paper.”)

I argue for accepting this paper as an Accept (poster)